# Presynaptic NMDARs cooperate with local spikes toward GABA release from the reciprocal olfactory bulb granule cell spine

Vanessa Lage-Rupprecht[1,2], Li Zhou[1], Gaia Bianchini[1], S Sara Aghvami[1,3,4], Max Mueller[1], Balázs Rózsa[5], Marco Sassoè-Pognetto[6], Veronica Egger[1]*

[1]Neurophysiology, Institute of Zoology, Universität Regensburg, Regensburg, Germany; [2]Department of Bioinformatics, Fraunhofer SCAI, Sankt Augustin, Germany; [3]School of Electrical and Computer Engineering, University of Tehran, Tehran, Islamic Republic of Iran; [4]School of Cognitive Sciences, Institute for Research in Fundamental Sciences (IPM), Tehran, Islamic Republic of Iran; [5]Two-Photon Imaging Center, Institute of Experimental Medicine, Hungarian Academy of Sciences, Budapest, Hungary; [6]Neuroscience Department, University of Turin, Torino, Italy

**Abstract** In the rodent olfactory bulb the smooth dendrites of the principal glutamatergic mitral cells (MCs) form reciprocal dendrodendritic synapses with large spines on GABAergic granule cells (GC), where unitary release of glutamate can trigger postsynaptic local activation of voltage-gated $Na^+$-channels ($Na_v$s), that is a spine spike. Can such single MC input evoke reciprocal release? We find that unitary-like activation via two-photon uncaging of glutamate causes GC spines to release GABA both synchronously and asynchronously onto MC dendrites. This release indeed requires activation of $Na_v$s and high-voltage-activated $Ca^{2+}$-channels (HVACCs), but also of NMDA receptors (NMDAR). Simulations show temporally overlapping HVACC- and NMDAR-mediated $Ca^{2+}$-currents during the spine spike, and ultrastructural data prove NMDAR presence within the GABAergic presynapse. This cooperative action of presynaptic NMDARs allows to implement synapse-specific, activity-dependent lateral inhibition, and thus could provide an efficient solution to combinatorial percept synthesis in a sensory system with many receptor channels.

*For correspondence:
Veronica.Egger@ur.de

## Introduction

Reciprocal dendrodendritic microcircuits can be found in several parts of the nervous system and are especially abundant in the vertebrate olfactory bulb (*Crespo et al., 2013*), where the dendrites of the principal mitral and tufted cells (MTCs) engage in such interactions with several major subtypes of local GABAergic neurons. In the glomerular layer, the MTC apical tuft is reciprocally connected to mostly periglomerular cell dendrites. In the external plexiform layer, the long MC lateral dendrites are densely covered with GABAergic synapses that mostly originate from reciprocal arrangements (*Bartel et al., 2015*; *Sailor et al., 2016*; *Matsuno et al., 2017*). While proximal circuits are thought to be mostly formed by granule cell (GC) spines (*Miyamichi et al., 2013*), there are also reciprocal dendrodendritic interactions with other GABAergic cell types such as SOM+ neurons, CRH+ neurons and most prominently parvalbumin/PV+ neurons that all feature aspiny, smooth dendrites (partially overlapping populations; *Toida et al., 1994*; *Lepousez et al., 2010*; *Huang et al., 2013*; *Kato et al., 2013*; *Miyamichi et al., 2013*).

Thus, the MC-GC synapse differs from the other bulbar reciprocal arrangements by its location within a large spine on the GC side. These spines feature particularly long necks (*Woolf et al., 1991*), which might serve to electrically isolate the synapse and boost local processing (e.g. *Miller et al., 1985*; *Spruston, 2008*). Indeed, we have gathered evidence that unitary postsynaptic potentials in GC spines are electrically compartmentalized and thus can locally activate voltage-gated Na$^+$-channels (Na$_v$s), which in turn activate classic presynaptic high-voltage-activated Ca$^{2+}$-channels (HVACCs) within the spine ('spine spike', *Bywalez et al., 2015*). Thus, the reciprocal spine might operate as a mini-neuron that can generate synaptic output upon local activation (*Egger and Urban, 2006*).

While there have been many earlier studies of recurrent dendrodendritic inhibition at the MTC-GC synapse, it is so far unknown whether a unitary, purely local activation of the spine's microcircuit can indeed trigger release of GABA back onto the exciting MTC. The issue is still unresolved for two reasons:

1. Most earlier studies used a strong stimulation of voltage-clamped MCs, namely a depolarization to 0 mV for 20–50 ms (*Isaacson and Strowbridge, 1998*; *Halabisky et al., 2000*; *Isaacson, 2001*; *Chen et al., 2000*). This protocol will cause massive release of glutamate from the lateral MC dendrites, invoking glutamate spillover between them (*Isaacson, 1999*) and resulting in long-lasting recurrent inhibition of MCs with both synchronous and asynchronous components, the latter with a time constant of ~ 500 ms. Under these circumstances and zero extracellular [Mg$^{2+}$]$_e$ GC NMDA receptors (NMDAR) were shown to provide postsynaptic Ca$^{2+}$ entry sufficient to trigger release of GABA from the GC spine (also in *Schoppa et al., 1998*). However, in normal [Mg$^{2+}$]$_e$ GABA release was mostly abolished by HVACC blockade (*Isaacson, 2001*; see Discussion).

2. In spite of these functional demonstrations of recurrent inhibition and of substantial ultrastructural evidence that GC spines onto MCs are usually reciprocal (e.g. *Price and Powell, 1970*; *Jackowski et al., 1978*; *Woolf et al., 1991*), functional connectivity between individual GCs and MCs has remained elusive: no reciprocal connectivity was reported from paired recordings of MCs coupled to GCs so far (*Isaacson, 2001*; *Kato et al., 2013*; *Pressler and Strowbridge, 2017*) and others including our group have been unable to find connected GC → MC pairs in the first place (see *Egger et al., 2003*; *Schoppa, 2006*). This puzzling observation indicates that paired recordings are not a suitable tool to investigate reciprocal release of GABA.

Here, we aim to determine whether – and possibly how - single inputs from MC lateral dendrites to single GC spines can also trigger recurrent release of GABA from the reciprocal synapse, using recordings from fluorescently labeled MCs and two-photon uncaging (TPU) of glutamate along their lateral dendrite near Venus-tagged GC spines. TPU of glutamate allows to study the operational mechanisms of the reciprocal microcircuit in single GC spines without causing broader GC activation (*Bywalez et al., 2015*, see also *Figure 1—figure supplement 1*). In this study, we use TPU at GC spines to investigate the reciprocal output by the GABAergic release machinery via recording of IPSCs in mitral cells (MCs). Thus, processing within a single spine can be disentangled in space and time from extended activation, in order to test the main predictions from our previous study: The mini-neuron hypothesis along with the GC spine spike phenomenon suggest that, just like in conventional axons, Na$_v$ and ensuing HVACC activation are required to cause release, and that unitary activation of the spine is sufficient for triggering release.

While this concept is validated here, our study also reveals that presynaptic NMDARs gate reciprocal output at the single-spine level. What is the advantage of such an arrangement compared to classical neuronal processing? In olfactory coding, a large number of input channels needs to be combined to synthesize an olfactory percept (e.g. *Mori et al., 1999*, *Murthy, 2011*). We propose that the mechanisms revealed here can explain the conspicuous apparent lack of functional reciprocal connectivity mentioned above and, more importantly, might allow reciprocal GC spines to play a central role in the efficient binding of changing sets of activated input channels and their glomerular columns.

## Results

### Experimental configuration

To enable pharmacological interference with components of the reciprocal microcircuit such as $Na_v$s and HVACCs, we bypassed release from the MC presynapse via TPU of DNI-caged glutamate (DNI, see Materials and methods; *Chiovini et al., 2014*; *Bywalez et al., 2015*; *Pálfi et al., 2018*) in acute juvenile rat olfactory bulb slices, while recording from MCs in somatic whole-cell voltage-clamp. To visualize the lateral dendrites, MCs were filled with the dye Alexa594 (50 µM). *Figure 1A* shows the recording configuration.

In all MC recordings we observed a high basal frequency of spontaneous events which is a general hallmark of the olfactory bulb network (*Egger and Urban, 2006*). Wash-in of 1 mM DNI further increased the basal frequency by on average 1.9 ± 1.2 times (mean value before DNI: 4.5 ± 2.1 Hz, n = 14 MCs, p<0.01, Wilcoxon test), due to the disinhibition resulting from a partial blockade of GABA$_A$Rs by DNI, which also reduced the mean spontaneous IPSC amplitude to a fraction of 0.47 ± 0.16 of control (n = 14 MCs, p<0.001, Wilcoxon test; *Figure 1—figure supplement 1A,B*). In the presence of DNI, the GABA$_A$R antagonist bicuculline (BCC, 50 µM) blocked spontaneous activity almost completely (frequency: 0.13 ± 0.10 of control, n = 6 MCs, p<0.01, Wilcoxon test; *Figure 1C*, *Figure 1—figure supplement 1A,B*).

Since in initial experiments we occasionally observed activation of both NMDA and AMPA autoreceptors on MC dendrites at a holding potential of −70 mV, MCs were clamped to +10 mV in the main body of experiments, near the reversal potential of ionotropic glutamate receptors (*Isaacson, 1999*; *Friedman and Strowbridge, 2000*). All experiments were performed at physiological levels of $[Mg^{2+}]_e$ (1 mM).

### Triggering of reciprocal IPSCs via TPU of glutamate

To prevent overstimulation with glutamate, we uncaged with similar laser parameter settings (see Materials and methods) as in the previous study, in which TPU-evoked GC $Ca^{2+}$ signals were indistinguishable from those of unitary spontaneous or triggered synaptic inputs, including their strict localization to the spine head (*Bywalez et al., 2015*), and routinely tested these settings; to ensure similar laser power across experiments, uncaging sites were located at a shallow depth no more than 20–30 µm below the surface of the tissue (see Materials and methods). In most experiments, uncaging was performed at one spot in the immediate vicinity of GC spines that were visible in VGAT-Venus rats and in close proximity (0.2–0.5 µm) to the MC lateral dendrite (e.g. *Figure 1A*, *Figure 2A*); in a few initial experiments in WT rats uncaging was performed 'blindly' at up to four spots along the dendrite to increase the likelihood for triggering a unitary response (e.g. *Figure 2B*; 18% of experiments, see Materials and methods). Responses were detectable in ~ 30% of tested MCs (total n = 166 MCs, see Materials and methods) and in this fraction of MCs would occur only in a small subset of stimulations (see below, *Figure 1F*), thus overstimulation of the circuit appears unlikely.

In addition, TPU-evoked signals can be expected to be localized to the stimulated GC spine: there is no $Ca^{2+}$ influx into the adjacent dendritic shaft upon TPU, as we have shown in GC $Ca^{2+}$ imaging experiments that were performed previously (dye 100 µM OGB-1, average amplitude ratio ΔF/F spine:dendrite 46:2 *Bywalez et al., 2015*) and also interleaved with the MC recordings in this study, using a low-affinity dye (100 µM OGB-6F, ratio ΔF/F spine:dendrite 24:2, *Ona Jodar et al., 2020*). Finally, we also tested via $Ca^{2+}$ imaging whether neighboring GC spines of the same GC could sense uncaged glutamate (n = 11 spine pairs, average distance between stimulated and non-stimulated spine heads 4.1 ± 2.2 µm, range 1.5–6.8 µm, *Figure 1—figure supplement 1C,D,E*). While the mean amplitude $(ΔF/F)_{TPU}$ in the stimulated spine set $(ΔF/F)_{TPU}$ = 52 ± 22% was similar to the data set in *Bywalez et al., 2015* (p=0.38, Mann-Whitney test), there were no detectable $Ca^{2+}$ transients in the non-stimulated spines (amplitude 2 ± 2%, p<0.002 vs stimulated spine, Wilcoxon test), and again there was no detectable $Ca^{2+}$ transient in the dendrite in between (mean 1 ± 2%, p=0.14 vs 0, Wilcoxon test; *Figure 1—figure supplement 1C,D*). All three compartments (stimulated and non-stimulated spine heads, dendritic shaft) showed similar $Ca^{2+}$ transients in response to somatically evoked action potentials (*Figure 1—figure supplement 1D*). There was no detectable influence of spine head distance on the relative difference in signal size between stimulated and

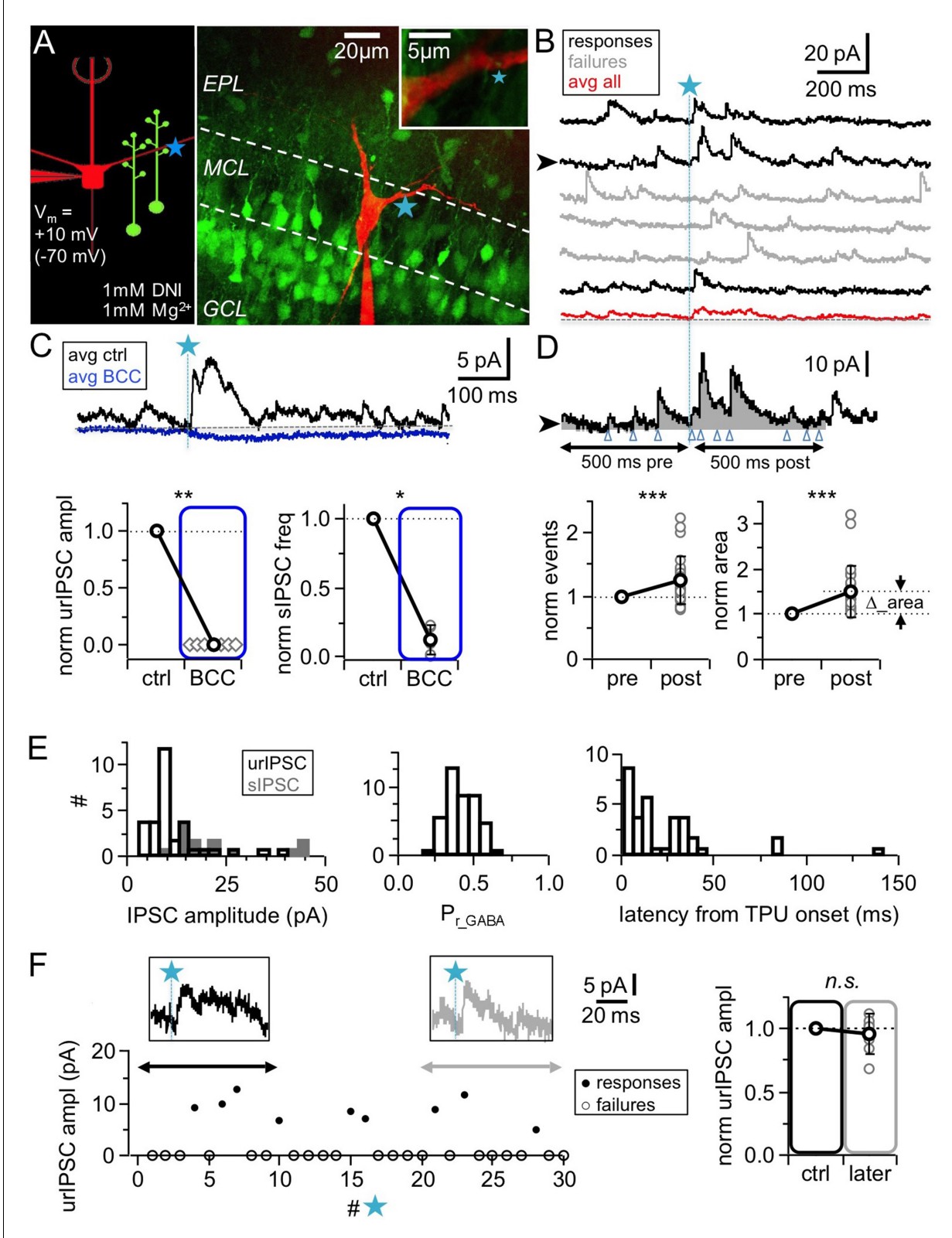

**Figure 1.** TPU-induced glutamatergic activation of GC spines triggers GABA release detected as uncaging-evoked reciprocal IPSCs (urIPSCs) in MCs. (**A**) Experimental setting (scheme and image): Somatic whole-cell voltage-clamp recording from MC filled with Alexa 594 (red, 50 µM) and TPU of DNI-caged glutamate (DNI; blue star) at GC spines in VGAT-Venus rats (GCs green). (**B**) Example of consecutive uncaging traces showing urIPSC responses (black) and failures (gray); average of all recordings shown below (red). (**C**) Effect of GABAₐR blockade (bicuculline, BCC, 50 µM, blue) on urIPSCs and

*Figure 1 continued on next page*

*Figure 1 continued*

spontaneous activity. Top: Example experiment, showing averaged traces. Bottom: Cumulative data on urIPSC amplitude (left panel, n = 7 MCs) and on frequency of spontaneous IPSCs (right panel, n = 6 MCs). (**D**) Analysis of asynchronous events. Top: Example trace from (**B**) with analysis intervals pre- and post-uncaging (gray areas, black arrows). Counted events marked by blue arrowheads. Bottom: Normalized IPSC event number and area (integrated activity, relative increase vs control Δ) in pre- vs post-uncaging intervals (n = 27 MCs, see Materials and methods). (**E**) Properties of first detected triggered urIPSCs (see Materials and methods). Left: Amplitudes (n = 32 MCs, $V_m$ = +10 mV). Dark gray: Amplitude distribution of spontaneous IPSCs for comparison (n = 14 MCs, mean amplitudes). Middle: Release probabilities $P_{r\_GABA}$ (n = 44 MCs). Right: Latencies from TPU onset (n = 36 MCs). (**F**) Stability of urIPSC recordings. Left: Representative experiment showing individual urIPSC amplitude values over consecutive stimulations (1 TPU per min). Insets: Averaged urIPSC responses in the first (black, n = 3 responses) and last ten minutes (gray, n = 3 responses). Right: Comparison of averaged normalized urIPSC amplitudes separated by 10 min interval (n = 7 MCs).

The online version of this article includes the following figure supplement(s) for figure 1:

**Figure supplement 1.** Effects of DNI (1 mM) on sIPSCs, spine-specificity of TPU, TPU sites, duration of responses.

---

non-stimulated spines (*Figure 1—figure supplement 1E*). Thus in our paradigm TPU can be expected to act in a spine-specific manner.

All uncaging spots were located proximally, at distances < 50 µm from the MC soma (*Figure 1—figure supplement 1F*). The rationale for this choice was to minimize electrotonic attenuation, since IPSC amplitudes were small (see above). Moreover, proximal stimulation and the use of VGAT-Venus rats should prevent inadvertent stimulation of reciprocal synapses with other interneuron types (see Introduction).

TPU of glutamate resulted in consecutive triggered responses and failures in MCs (*Figure 1B*). Responses were classified as triggered if they were observed repeatedly within the same time window and showed similar kinetics (see Materials and methods). Next, we tested whether these triggered responses were indeed GABAergic by applying the $GABA_AR$ antagonist bicuculline (BCC, 50 µM), which invariably blocked responses completely (*Figure 1C*, n = 7 MCs, p<0.01, Wilcoxon test). Thus in the following the triggered events are denoted as urIPSCs (**u**ncaging-evoked **r**eciprocal IPSCs).

The amplitude of triggered urIPSCs was on average 12 ± 8 pA (in n = 32 MCs clamped to +10 mV, *Figure 1E*), which is significantly smaller than the mean amplitudes of spontaneous IPSCs (sIPSCs, n = 14 MCs; 22 ± 12 pA; p<0.001, Mann-Whitney test, *Figure 1E*, *Figure 1—figure supplement 1B*). Since sIPSCs are highly likely to mainly originate from the reciprocal MC-GC circuit (see Discussion), this observation further argues against overstimulation with glutamate in our experiment.

The average release probability from the reciprocal spine was estimated as $P_{r\_GABA}$ = 0.34 ± 0.11 (*Figure 1E*, range 0.13–0.60, based on n = 44 MCs, see Materials and methods and Discussion). The latencies of urIPSCs were not normally distributed (*Figure 1E*), with a first peak within the first 10 ms post TPU onset (n = 13 MCs), a second peak around 30 ms (n = 19 MCs within the range 10–50 ms) and a yet more delayed tail (n = 3 MCs, see *Figure 1—figure supplement 1H* and Discussion).

In most experiments we detected putative asynchronous urIPSCs following the first triggered event which were quantified via integral analysis and counting of events (*Figure 1D*, see Materials and methods and Discussion). Both area and number of events increased highly significantly in the 500 ms interval following TPU ('post') compared to the same interval right before ('pre'; mean increase of integrated activity Δ_area + 0.50 ± 0.55 relative to 'pre' value; mean increase in event number + 0.25 ± 0.37 relative to 'pre' value; n = 27 MCs, p<0.001 for both, Wilcoxon test; absolute values in 'pre' area 2.40 ± 1.83 pAs, event numbers 26.1 ± 14.9). Δ_area was also significantly increased if the extra area of the synchronous urIPSC was subtracted (p<0.005). The total duration of recurrent inhibition was on average 179 ± 137 ms (range 32–533 ms, n = 26 MCs, *Figure 1—figure supplement 1G*). Thus, asynchronous recurrent inhibition can be already triggered by single glutamatergic inputs.

Sufficient stability of urIPSC responses was established via long-term recordings (see Materials and methods). The amplitudes of averaged responses during the first and last 5–10 photostimulations (with at least 10 min in between to mimic the time course of pharmacological manipulations, mean number of responses per averaged urIPSC 2.8 ± 1.1) were not significantly different (*Figure 1F*, n = 7 MCs, ratio last to first 0.95 ± 0.15, p=0.74, Wilcoxon test).

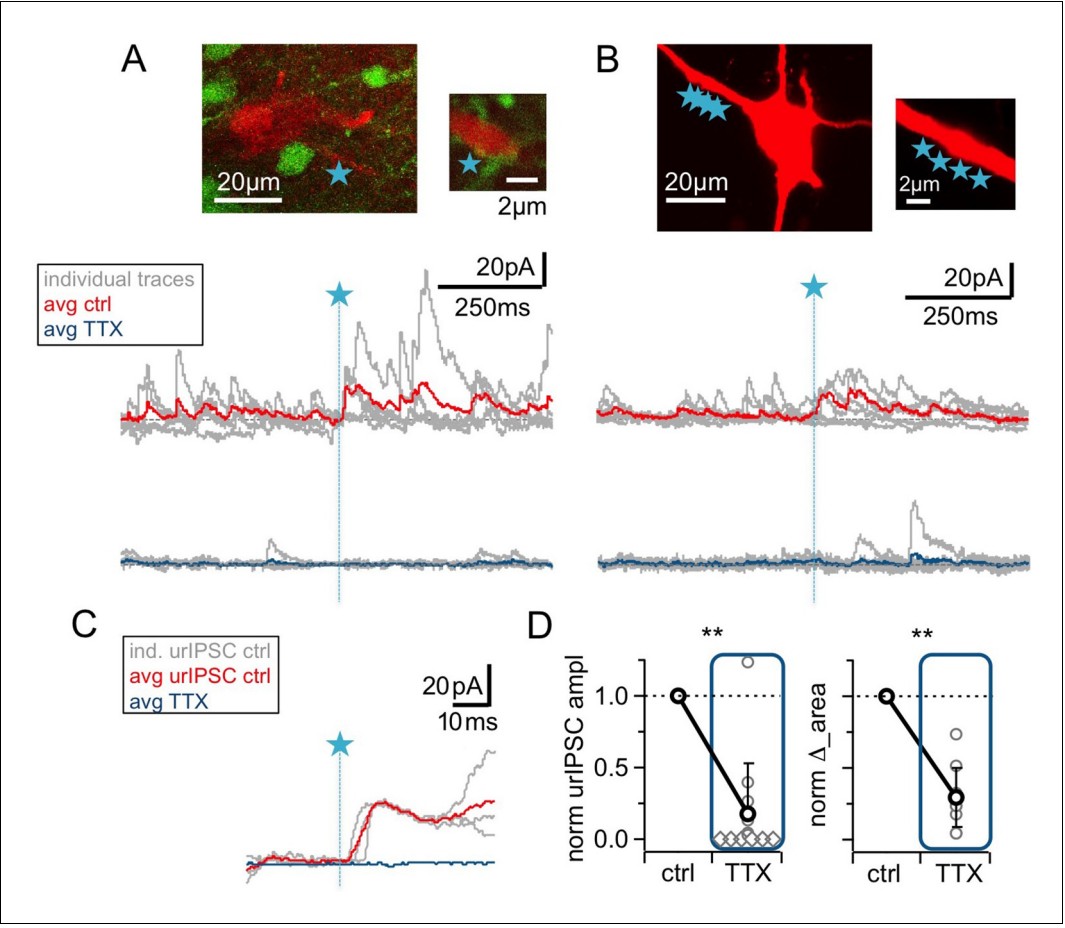

**Figure 2.** The urIPSC is reduced by Na$_v$ blockade (TTX, 500 nM). (A, B) Representative experiments showing a patch-clamped mitral cell (Alexa594 50 μM, red), the uncaging site(s) along a lateral dendrite (blue star) and below the corresponding uncaging traces with individual traces shown in gray, average control traces in red and average TTX traces indicated in blue (A: VGAT-Venus rat, single-site TPU, B: Wistar rat, multi-site TPU). The average traces contain all recordings, both with and without detected responses. (C) Magnified illustration of traces in (A) (for control individual traces with urIPSC responses and their average, for TTX only average, color coding as above). (D) Left: Cumulative normalized data showing the strong reduction of urIPSC amplitude during Na$_v$ blockade with TTX (n = 12 MCs). Diamonds indicate the experiments with no detectable response in the presence of the drug. Right: Comparison of normalized IPSC Δ_area between control and in the presence of TTX (see Materials and methods, n = 10 MCs).

The online version of this article includes the following figure supplement(s) for figure 2:

**Figure supplement 1.** Effects of Na$_v$ blockade (TTX, 500 nM) on spontaneous activity.

From all these experiments, we conclude that local, unitary-like TPU of glutamate can stably trigger both fast and slow reciprocal release of GABA, providing proof that the reciprocal microcircuit can be activated by single synaptic inputs at physiological levels of Mg$^{2+}$.

## Sodium channel activation is required for most urIPSCs

Next, we investigated a possible contribution of the GC spine spike and thus Na$_v$ activation to urIPSC generation. *Figure 2* illustrates that wash-in of TTX (500 nM) substantially reduced both triggered and spontaneous events. On average, urIPSC amplitudes were reduced to a fraction of 0.17 ± 0.34 of control (*Figure 2D*, n = 12 MCs, p<0.005, Cohen's d = 2.54, Wilcoxon test; absolute control amplitudes for −70 mV: −7.8 ± 3.6 pA, n = 5 MCs; for +10 mV: 15.4 ± 12.4 pA, n = 7 MCs), and the increase in integrated activity Δ_area was decreased to 0.28 ± 0.21 of its control value (see Materials and methods; *Figure 2D*, n = 10 MCs, p<0.005).

While the frequency of spontaneous activity was also strongly decreased in TTX, to a fraction of 0.24 ± 0.17 of control (*Figure 2—figure supplement 1B*, n = 10 MCs, p<0.005), the mean sIPSC amplitude was unchanged by TTX (1.36 ± 0.68 of control, n.s., n = 10 MCs, *Figure 2A,B*, *Figure 2—figure supplement 1A,B*). There was no significant correlation between the TTX-induced reductions in sIPSC frequency and in urIPSC amplitude (*Figure 2—figure supplement 1C*, n = 10, r = 0.23, p=0.26), and therefore the reduction in urIPSC amplitude is unlikely to be directly related to network effects of TTX.

Thus Na$_v$s are essential to trigger GABA release from the reciprocal spines.

## High-voltage activated Ca$^{2+}$ channels in the spine trigger GABA release

HVACCs have been implied to mediate recurrent release from reciprocal spines (*Isaacson, 2001*) and are activated by Na$_v$s, contributing a substantial fraction to the total postsynaptic Ca$^{2+}$ signal in the GC spine (*Bywalez et al., 2015*). To directly test whether HVACC activation is required for release of GABA, we blocked NPQ-type Ca$^{2+}$ channels with 1 µM ω-conotoxin MVIIC (CTX; *Bloodgood and Sabatini, 2007*; *Bywalez et al., 2015*).

*Figure 3* shows the resulting substantial decrease of urIPSCs, to a fraction of 0.08 ± 0.14 of control (from a mean amplitude of 11.3 ± 5.7 pA; n = 8 MCs, p<0.005, Cohen's d = 6.6, *Figure 3D*). This decrease was not different from the effect of TTX on urIPSC amplitude described above (p=0.35, ratios in CTX vs TTX). Δ_area decreased to 0.39 ± 0.23 of control (n = 9 MCs, p<0.005, *Figure 3D*), again statistically not different from the effect of TTX on Δ_area described above (p=0.15 vs TTX).

CTX decreased sIPSC frequency to a fraction of 0.53 ± 0.15 of control, less markedly than TTX (*Figure 3E*, n = 7 MCs, control vs CTX: p<0.01; CTX vs TTX: p<0.005; Mann-Whitney test), and had no effect on the mean sIPSC amplitude (*Figure 3A,B*; *Figure 3—figure supplement 1A,B*; 1.00 ± 0.12 of control, n = 7 MCs, n.s.). There was no significant correlation between the CTX-induced reductions in sIPSC frequency and in urIPSC amplitude (*Figure 3—figure supplement 1C*, n = 7, r = −0.38, p=0.19). Therefore, the reduction in urIPSC amplitude is unlikely to be directly related to network effects of CTX.

We conclude that HVACC activation is also required for release of GABA from the reciprocal spine following local input.

## NMDA receptors are also relevant for recurrent release

NMDARs are known to substantially contribute to unitary synaptic transmission and to postsynaptic Ca$^{2+}$ entry at the MC to GC synapse (*Isaacson and Strowbridge, 1998*; *Schoppa et al., 1998*; *Isaacson, 2001*; *Egger et al., 2005*). However, because of our previous observation that NMDAR-mediated Ca$^{2+}$ entry into GC spines did not depend on Na$_v$ activation (in contrast to HVACC-mediated Ca$^{2+}$ entry, *Bywalez et al., 2015*), and because of the strong blocking effects of TTX or CTX on urIPSCs reported above, we expected that NMDAR blockade would have only mild effects on fast recurrent release upon single inputs. Intriguingly however, *Figure 4A–D* shows that the application of 25 µM D-APV resulted in a substantial decrease of urIPSC amplitudes, to on average 0.22 ± 0.21 of control (from a mean amplitude of 13.8 ± 8.6 pA, n = 10 MCs, p<0.002, Cohen's d = 3.7, Wilcoxon test; *Figure 4D*). All individual experiments showed the amplitude decrease, albeit to a variable degree (range 0.00–0.68 of control). Δ_area following TPU decreased to 0.40 ± 0.28 of control (n = 10 MCs, p<0.003, *Figure 4D*). Both effects were statistically not different from the effects of either TTX or CTX (amplitude: p=0.12 and p=0.08; Δ_area: p=0.19 and p=0.47; Mann-Whitney test).

While APV also substantially reduced sIPSC frequency, to 0.27 ± 0.23 of control (n = 10 MCs, p<0.005, *Figure 4A,B*; *Figure 4—figure supplement 1A,B*), similar to TTX (p=0.79) and significantly more pronounced than CTX (p<0.01), the mean sIPSC amplitude was unchanged (*Figure 4A,B*; *Figure 4—figure supplement 1A,B*; 0.93 ± 0.29 of control, n = 10 MCs, p=0.19). There was no positive correlation between the APV-induced reductions in sIPSC frequency and urIPSC amplitude (*Figure 4—figure supplement 1C*, n = 10, r = −0.23, p=0.26). Therefore network effects of APV are unlikely to explain the strong reduction of urIPSC amplitude.

Since the strength of the effect of NMDAR blockade on urIPSCs was surprising to us, we sought to provide another line of evidence for these findings in an experimental setting that does not

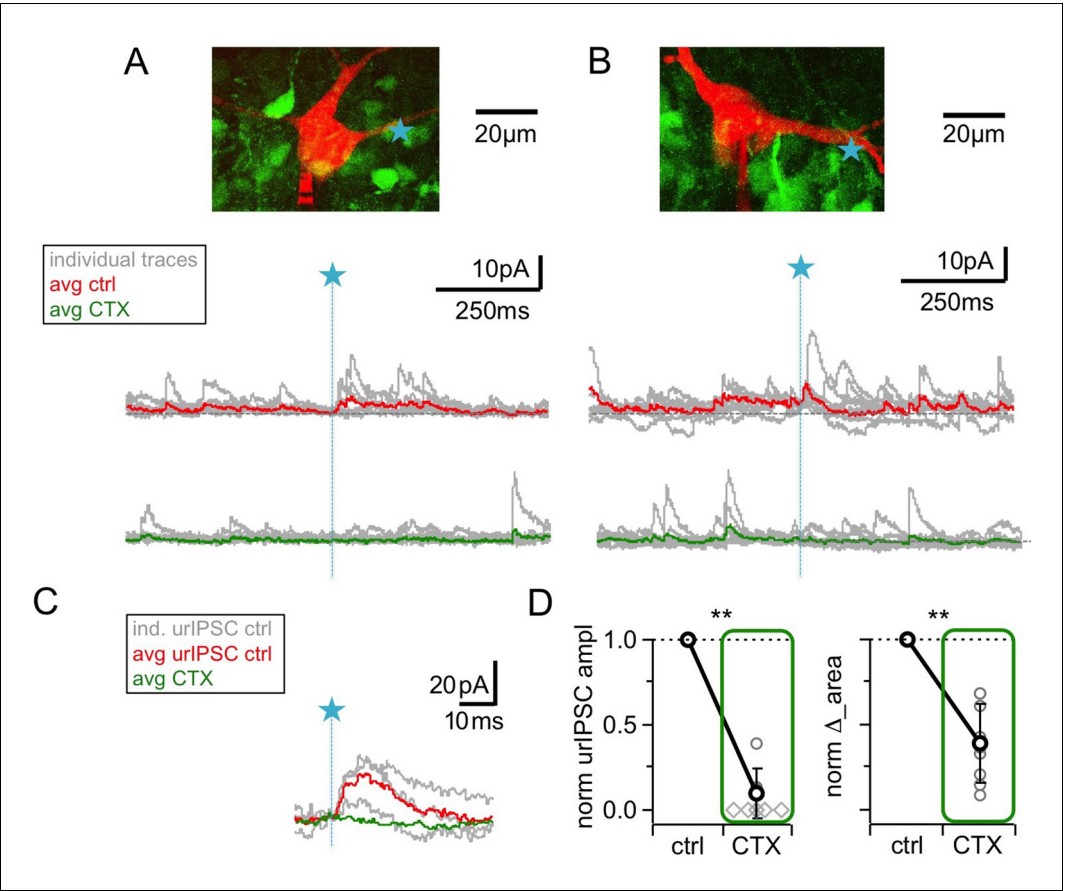

**Figure 3.** Blockade of high voltage activated Ca$^{2+}$-channels by ω-conotoxin MVIIC (CTX, 1 μM) causes a prominent reduction of urIPSC amplitudes. (**A, B**) Two representative experiments in brain slices from VGAT-Venus rat with the corresponding MC (red), the site of TPU (blue star) and the uncaging traces according to the condition (individual traces: gray, average control: red, average CTX: green). The average traces contain all recordings, both with and without detected responses. Left: same cell as in **Figure 1B**. (**C**) Magnified illustration of traces in B (for control individual traces with urIPSC responses and their average, for CTX only average, color coding as above). (**D**) Left: Summary of effects of CTX on average normalized urIPSC amplitude (n = 8 MCs). Diamonds indicate the experiments with no detectable response in the presence of the drug. Right: Comparison of delta urIPSC areas normalized to control vs in the presence of CTX (n = 9 MCs).

The online version of this article includes the following figure supplement(s) for figure 3:

**Figure supplement 1.** Effects of HVACC blockade by ω-conotoxin MVIIC (CTX, 1 μM) on spontaneous activity.

---

involve uncaging of glutamate (which might preferentially activate NMDARs, see Discussion). The afterhyperpolarization (AHP) following single MC APs elicited by somatic current injection was found to mainly reflect recurrent inhibition in brain slices from adult mouse (**Nunes and Kuner, 2018**), while in juvenile rat inhibitory synaptic contributions to the AHP were observed to be less substantial but still detectable (on the order of 20–30% relative to the K$^+$ channel-mediated component, **Duménieu et al., 2015**). We used this paradigm to test whether NMDAR blockade alone could interfere with recurrent inhibition (**Figure 4E,F**). Single MC AP AHPs (V$_{hold}$ = − 70 mV) had a mean amplitude $\Delta V_m$ = − 9.2 ± 2.2 mV (n = 18) and a mean half duration $\tau_{1/2}$ = 43 ± 25 ms (n = 13). **Figure 4F** shows that APV application significantly reduced the mean AHP amplitude to a fraction of 0.86 ± 0.11 of control (p<0.0001, Wilcoxon test, Cohen's d = 0.86; reduction in 17 of 18 MCs), while the half duration was not changed (fraction of control 1.02 ± 0.42, p=0.65, not shown).

Next, to block GABA$_A$Rs we turned to gabazine (GBZ, 20 μM), since bicuculline might affect the K$^+$ channels that contribute substantially to slow AHPs (**Khawaled et al., 1999**; **Duménieu et al., 2015**). GBZ alone caused a slightly stronger reduction than APV, which was not significantly different (to 0.71 ± 0.10 of control, n = 7 MCs, p=0.08 vs APV, Mann-Whitney test, not shown). Occlusion

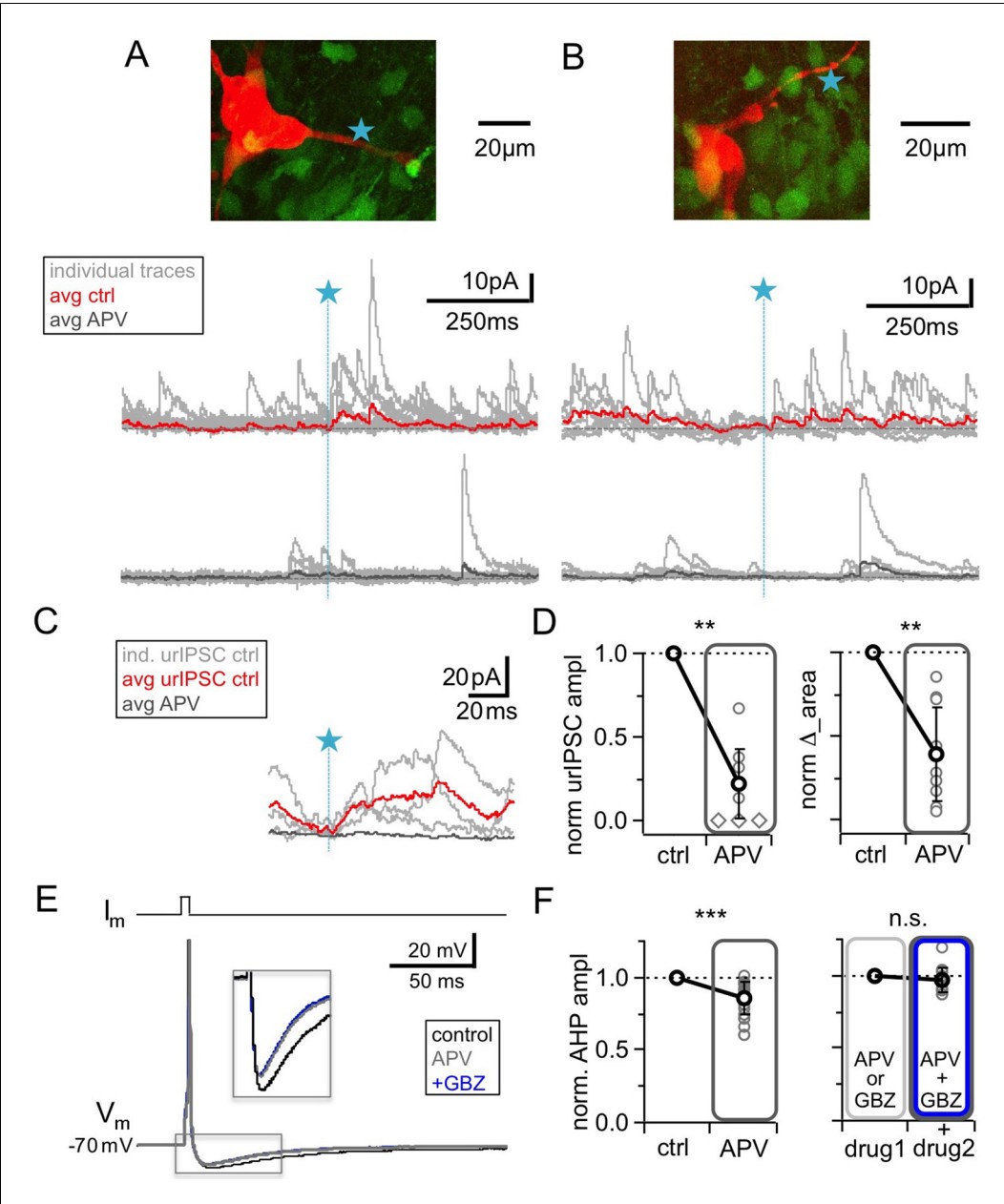

**Figure 4.** NMDAR blockade with D-APV (25 µM) results in a strong reduction of urIPSC amplitudes and also reduces AHPs following mitral cell APs. (**A, B**) Two representative uncaging experiments with the corresponding MC (red), the site of TPU (blue star) and the uncaging traces according to the condition (individual traces: gray, average control: red, average APV: dark gray); both VGAT-Venus. The average traces contain all recordings, both with and without detected responses. (**C**) Magnified illustration of traces in A (for control individual traces with urIPSC responses (gray) and their average (red), for APV only average (dark gray)). (**D**) Left: Summary of effects of APV on average normalized urIPSC amplitude (n = 10 MCs). Diamonds indicate the experiments with no detectable response in the presence of the drug. Right: Comparison of delta urIPSC integrals between control vs in the presence of APV (n = 10 MCs). (**E**) Representative example of MC AP evoked by somatic current injection in control conditions (black trace) and in the presence of APV (gray trace) and added GBZ (20 µM; blue trace). Inset: Magnified afterhyperpolarizations (AHPs). (**F**) Left: Cumulative effects of APV on average normalized AHP amplitude (n = 18 MCs). Right: Effect of occlusion experiments on AHP amplitude (APV before GBZ or GBZ before APV, n = 12 MCs).

The online version of this article includes the following figure supplement(s) for figure 4:

**Figure supplement 1.** Effects of NMDAR blockade by D-APV (APV, 25 µM) on spontaneous activity.

experiments using APV before GBZ or GBZ before APV did not result in any further reduction (0.97 ± 0.09 of amplitude in the presence of first drug, n = 12 MCs, p=0.16, Wilcoxon test, *Figure 4E,F*). Thus these experiments also support a substantial role of GC NMDARs for GABA release.

From these and the uncaging experiments we conclude that NMDARs located on GCs are strongly involved in reciprocal release, even though prevention of HVACC activation (via $Na_v$ block or pharmacology) also blocks reciprocal release. What is the underlying mechanism?

The simplest explanation for a cooperation would be a summation of NMDAR- and HVACC-mediated $Ca^{2+}$ currents at the presynaptic $Ca^{2+}$ sensor. For this scenario two requirements need to be satisfied: (1) temporal overlap of $Ca^{2+}$ currents and (2) spatial proximity of NMDARs and HVACCs within the same microdomain.

## Temporal overlap: simulations of GC spine $Ca^{2+}$ currents via HVACCs and NMDARs

In conventional glutamatergic synapses the NMDAR-mediated postsynaptic current is rising rather slowly compared to the AMPAR-mediated component (rise time ~ 10 ms, *Lester et al., 1990*) and therefore the fractional NMDAR $Ca^{2+}$ current seems at first an unlikely candidate to make direct contributions to fast release. In the reciprocal GC spines however, $Na_v$ channels are locally activated, which enables fast and substantial NMDAR activation. *Figure 5A* shows a simulation of the postsynaptic NMDAR- and HVACC-mediated $Ca^{2+}$ currents (based on the detailed NEURON model described in *Aghvami et al., 2019*, see Materials and methods) which illustrates that the local AP causes a transient boosting of the NMDAR $Ca^{2+}$ current because of further relief of the $Mg^{2+}$ block during the upstroke and overshoot of the spine spike. According to the simulation this fast NMDAR-mediated $Ca^{2+}$ spikelet begins even before the HVACC-mediated current and tightly overlaps with it within >1 ms. Thus HVACC- and NMDAR-mediated $Ca^{2+}$ currents could act in a cooperative manner at the $Ca^{2+}$ sensor(s) that mediate fast release of GABA, especially so, if the release probability was proportional to the fourth power of local $\Delta[Ca^{2+}]$ or more as at other central synapses, and if the channels were close enough to form a microdomain, allowing for an 'overlap bonus' (*Stanley, 2016*; *Figure 5B*).

In the temporal domain, this overlap was found to be highly robust against the variation of the combined $Na_v/K_v$ conductance, and increases in either AMPAR conductance $g_{AMPA}$ or neck resistance $R_{neck}$ resulted in an earlier activation of both HVACC and NMDAR-mediated $Ca^{2+}$ currents and even stronger overlap (*Figure 5C*, see Materials and methods). Decreases resulted at first in little change and then a loss of the spine spike whereupon there is no more HVACC activation. To illustrate specifically the contribution of the $Na_v$-mediated boosting of the NMDAR-mediated $Ca^{2+}$ current, we also calculated the overlap bonus for the HVACC-mediated current as above and the NMDAR current in absence of the spine spike, which indeed renders the cooperative effect negligible (*Figure 5B*).

## Spatial proximity: ultrastructural evidence for presynaptic localization of NMDARs

For fast neurotransmitter release to occur, the current view is that $Ca^{2+}$ entry needs to happen in a proximity of 100 nm or less from the $Ca^{2+}$ sensing part of the SNARE machinery (reviewed in *Kaeser and Regehr, 2014*; *Stanley, 2016*). Therefore, to permit cooperation of $Ca^{2+}$ influxes, NMDARs should be localized very close to or within the GABAergic presynapse. Earlier electron microscopic (EM) studies had reported the presence of postsynaptic NMDARs in GC spines and also instances of extrasynaptic labeling (*Sassoè-Pognetto and Ottersen, 2000*; *Sassoè-Pognetto et al., 2003*). To test for the presence of NMDARs near or within GABAergic active zones, we analyzed GC spine heads in ultrathin sections that were immunogold-labeled with either GluN1 or GluN2A/B antibodies (*Figure 6A*; see Materials and methods); the sections were selected for the presence of at least one asymmetric (i.e. glutamatergic) contact and/or one symmetric (i.e. GABAergic) contact.

Interestingly, we observed that when both symmetric and asymmetric synaptic profiles were visible in individual GC-MC dendrodendritic pairs, the two synaptic profiles were mostly either contiguous (44% of n = 60 cases, examples *Figure 6A2, A3*, cumulative plot *Figure 6B*) or closer than 100 nm (36% of cases). This finding implies that on both sides of the dendrodendritic synaptic

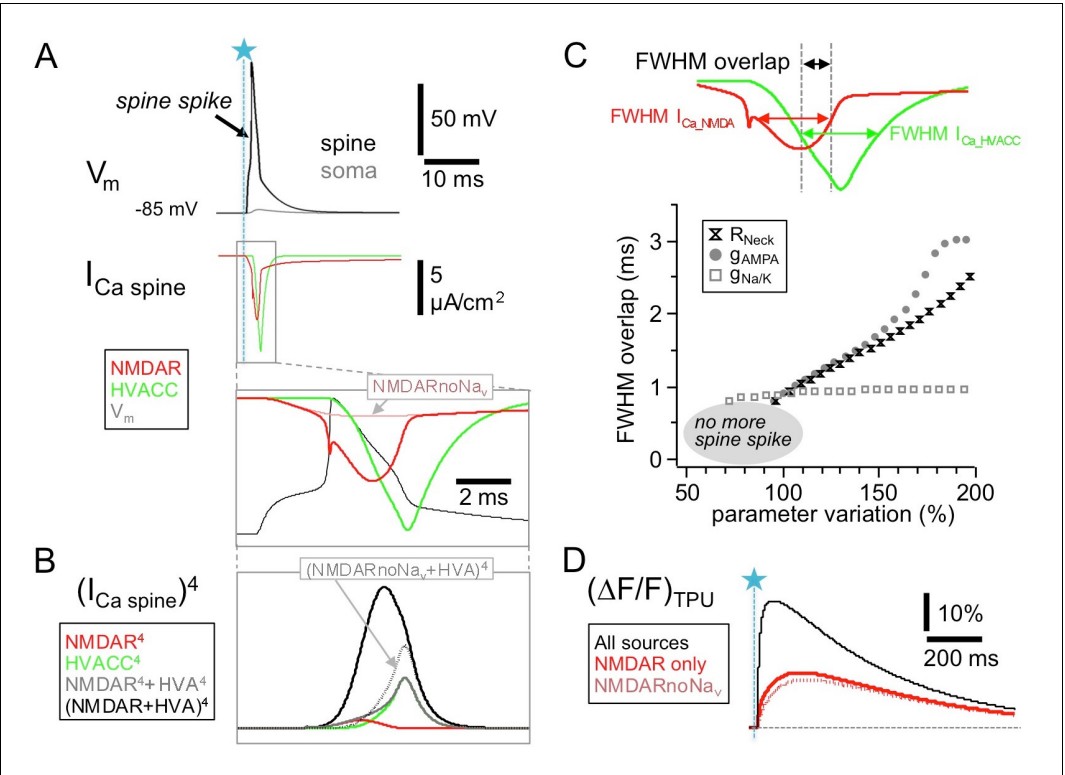

**Figure 5.** HVACC- and NMDAR-mediated Ca$^{2+}$-currents overlap temporally in the wake of the spine spike. Simulated spine Ca$^{2+}$-currents and ($\Delta$F/F)$_{TPU}$. (**A**) Top panels: spine membrane potential V$_m$ and Ca$^{2+}$-currents (parameters exactly as in *Aghvami et al., 2019*, their Figure 2B, except for removed exogenous Ca$^{2+}$ buffer). Bottom: Blow-up of NMDAR- and HVACC-mediated Ca$^{2+}$-currents, overlaid with the membrane potential V$_m$. Note the early onset of the NMDAR Ca$^{2+}$-current and its increase during the underlying local action potential, with a notch caused by the reduction of driving force during the AP upstroke. Red dotted line: NMDAR Ca$^{2+}$-current in the absence of Na$_v$s. (**B**) The temporal coincidence creates an 'overlap bonus' with respect to local Ca$^{2+}$ concentration and triggering of release: Fourth power of single and added Ca$^{2+}$-currents. Black dotted line: Addition of HVACC with NMDAR Ca$^{2+}$-current in the absence of Na$_v$s (purely hypothetical: pretending that HVACC was not affected, only the NMDAR Ca$^{2+}$-current). (**C**) Robustness test for extent of overlap between NMDAR- and HVACC-mediated Ca$^{2+}$-current. Overlap is measured as the stretch of overlapping FWHMs of the current transients (Full Width Half Maxima, see Materials and methods). Parameter variation within 50–200% of the nominal value of the neck resistance R$_{neck}$, the AMPAR conductance gAMPA and the coupled Na$_v$/K$_v$ conductance g$_{Na/K}$. Results are shown only for parameter runs with spine spike. (**D**) Simulated fluorescence transients $\Delta$F/F (in the presence of 100 µM OGB-1) show that the contribution of the Na$_v$-mediated boost of NMDAR Ca$^{2+}$-currents to the total NMDAR-mediated Ca$^{2+}$ signal is almost negligible (see Discussion).

arrangement the postsynaptic specialization very often merges with presynaptic release sites without interposition of non-synaptic membrane domains, and that therefore on the GC side postsynaptic NMDARs can be located very closely to the GABAergic presynapse.

Immunogold labeling for NMDARs was also frequently observed in symmetric contacts, albeit at a lesser density compared to asymmetric contacts (*Figure 6A*). To establish whether there is an increased likelihood for the presence of NMDARs at symmetric contacts vs extrasynaptic profiles, we compared the densities of labeling in non-synaptic membrane segments of GC spines with those in symmetric synapses, and as a control in asymmetric synapses. GC spines establishing synapses with MC dendrites were selected as described (see Materials and methods) and all discernible membrane segments within a spine were analyzed. For GluN1, there were 50 particles along 84.5 µm of 138 non-synaptic spine membrane segments (mean 0.6 particles/µm), 70 particles along 32.7 µm of 120 symmetric profiles (mean 2.1 particles/µm) and 394 particles along 28.6 µm of 111 asymmetric profiles (mean 13.8 particles/µm). *Figure 6C* shows the density distributions across the three types of membrane segments. The distribution of the density of gold particles at symmetric synapses was

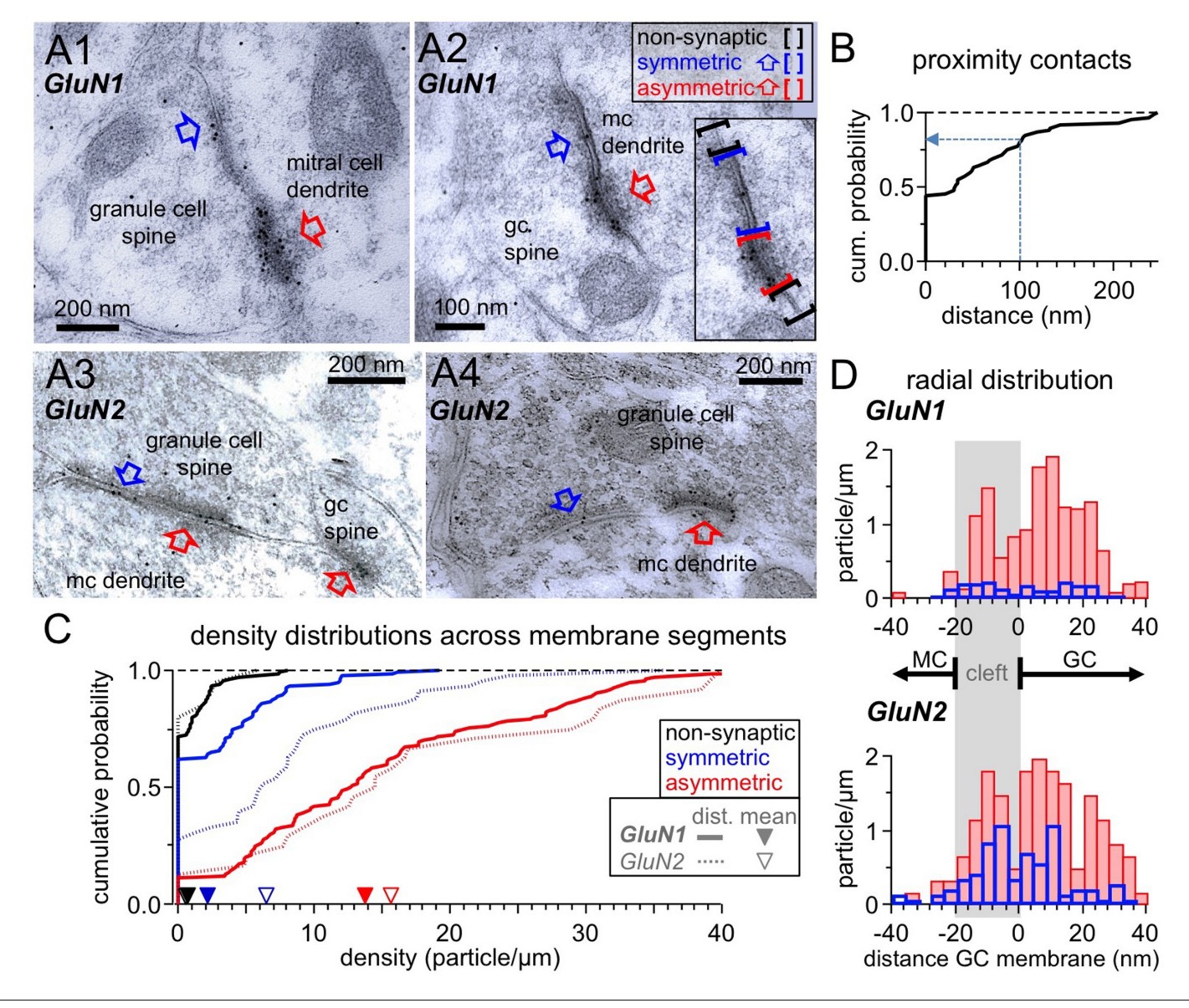

**Figure 6.** Ultrastructural analysis of the distribution of NMDARs shows their presence at both post- and pre-synaptic GC spine membranes in reciprocal dendro-dendritic synapses. (**A**) Representative micrographs of reciprocal dendrodendritic synapses labeled for GluN1 (A1, A2) and GluN2A/B (A3, A4). Note that asymmetric MC-to-GC synapses (red arrows) are strongly labeled and that symmetric GC-to-MC junctions (blue arrows) are also immunopositive. Gold particles are usually not associated with non-synaptic plasma membrane domains of GC spines. The inset in panel A2 illustrates the selection of non-synaptic membrane segments (black brackets), symmetric synapses (blue) and asymmetric synapses (red). (**B**) Frequency distribution of distances between symmetric and asymmetric synaptic profiles on the same GC spine membrane (based on n = 60 micrographs with both types of profiles on the same spine). (**C**) Frequency distributions of labeling densities across synaptic (symmetric and asymmetric) and non-synaptic membrane segments for GluN1 (thick lines, n = 120, 111, 138 segments, respectively) and GluN2A/B (thin dotted lines, n = 46, 25, 24 segments, respectively). The arrows on the x-axis indicate the mean density (GluN1: solid, GluN2A/B: open). Color code as in (**A**). (**D**) Radial distribution of gold particles representing GluN1 (top panel) and GluN2A/B (bottom panel) at asymmetric (red) and symmetric (blue) synapses. Particle numbers were normalized to the total length of the respective synaptic segments. Absolute particle numbers see Results.

significantly higher than the one in non-synaptic membrane (p=0.000029, Kolmogorov-Smirnov test). For GluN2A/B, the sample size was smaller, with 5 particles along 10.0 µm of 24 non-synaptic spine membrane segments (mean 0.5 particles/µm), 89 particles along 14.0 µm of 46 symmetric profiles (mean 6.4 particles/µm) and 96 particles along 6.1 µm of 25 asymmetric profiles (mean 15.7

particles/μm). Still, the density distribution at symmetric synapses was significantly higher than the distribution at the non-synaptic membrane (p=0.000048).

Interestingly, while the labeling densities for GluN1 and GluN2A/B were similar across non-synaptic membranes and asymmetric synapses each (p=0.99 and p=0.81, respectively), labeling of symmetric synapses was considerably stronger for GluN2A/B vs GluN1 (p=0.00042). This difference might be due to a distinct receptor configuration and/or a better accessibility of the GluN2 epitopes in the presynaptic symmetric membrane.

Finally, we analyzed the radial distribution of particles across synaptic profiles for both antibodies to establish whether labeling was predominantly associated with either the GC or the MC membrane (*Figure 6D*; see Materials and methods). The distribution of both GluN1 and GluN2A/B label at asymmetric synapses was similar, with a main peak just inside the postsynaptic GC membrane and another peak within the synaptic cleft. Only a few gold particles were localized beyond 20 nm on the MC side, suggesting that NMDARs are not expressed at significant levels at the glutamate release sites on the MC membrane. Notably, the distribution of gold particles at symmetric synapses mirrored the one at asymmetric synapses, implying that labeling for NMDARs is predominantly associated with both the presynaptic GABAergic and the postsynaptic glutamatergic membrane of GCs.

In summary, we conclude from the combined physiological, computational and ultrastructural evidence that NMDARs do play a direct presynaptic role for GABA release from the reciprocal spine, albeit in cooperation with the spine spike, and propose that this cooperation could be mediated by overlapping NMDAR- and HVACC-mediated $Ca^{2+}$ entry (*Figure 7*). A direct consequence of this conclusion is that any inhibitory output from a GC spine will require the presence of glutamate and thus synaptic input to the respective spine (except perhaps for GC action potential bursts), with far-ranging implications for the role of GC-mediated lateral inhibition in early olfactory processing (see Discussion).

## Discussion

Here we have demonstrated that mimicking unitary synaptic inputs via two-photon uncaging of glutamate onto individual olfactory bulb GC spines can activate the entire microcircuit within the spine, from the local spine spike to the release of GABA onto MC lateral dendrites, proving the mini-neuron-like functionality of the reciprocal microcircuit. As in classical axonal release, sequential $Na_v$ channel and HVACC activation triggers output, which occurs on both fast and slow time scales. Strikingly, however, presynaptic NMDA receptors are also found to play a role for GABA release. These findings together with other prior knowledge allow to make an educated guess regarding the specific function of reciprocal spines in bulbar processing (see further below).

### Properties of microcircuit operation and implications for coincident local and global activation

For the proximal reciprocal GC inputs investigated here we estimate that under physiological conditions close to the MC resting potential the size of the fast IPSCs is on the order of −5 pA, after corrections for the partial GABA$_A$R block by DNI and the setting of $E_{Cl}$. Thus, assuming an in vivo MC input resistance of 100 MΩ (*Angelo and Margrie, 2011*), an inhibitory input from a single GC spine will exert a somatic hyperpolarization of at best 0.5 mV, and therefore even proximal GC inputs will barely influence MC firing (*Fukunaga et al., 2014*; *McIntyre and Cleland, 2016*) - unless there is coordinated activity across GC spines connected to the same MC dendrite, for example in the wake of an MC action potential during the recurrent IPSP (see e.g. *Figure 4F*) or during gamma oscillations (e.g. *Kay, 2003*; *Lagier et al., 2004*).

Upon local activation we observed a GABA release probability $P_{r\_GABA}$ from the GC spine on the order of 0.3. This value might represent an upper limit, because the global reduction of inhibition by DNI could cause a homeostatic upregulation of $P_{r\_GABA}$ (e.g. *Rannals and Kapur, 2011*), and the detection of connections is generally biased toward larger $P_{r\_GABA}$. Moreover, $P_{r\_GABA}$ might become further downregulated during development, since recurrent inhibition was shown to decrease in older animals (*Dietz et al., 2011*; *Duménieu et al., 2015*).

With $P_{r\_GABA} \approx 0.3$ and the probability for MC glutamate release on the order of $P_{r\_Glu} \approx 0.5$–0.75 (*Egger et al., 2005*; *Pressler and Strowbridge, 2017*) the efficiency of the entire reciprocal microcircuit can be estimated as $P_{reciprocal} = P_{r\_Glu} \cdot P_{r\_GABA} \approx 0.2$, possibly informing future network

models. The rather low $P_{r\_GABA}$ observed here also implies that GC spines are likely to release with higher probabilities upon coincident non-local GC signaling (i.e. regional or global $Ca^{2+}$ or $Na^+$ spikes), due to substantially increased $\Delta Ca^{2+}$ in the spine as observed previously (*Egger et al., 2005*; *Egger, 2008*; *Aghvami et al., 2019*; *Mueller and Egger, 2020*). Therefore, we predict such coincident non-local activity to boost both recurrent and lateral inhibition.

As to the minimal latency for recurrent GABA release, the temporal resolution of our experiments is limited by the uncaging laser pulse duration (1 ms) and by the unknown exact time course $V_m(t)_{SPINE}$ of the spine spike. *Figure 1E* shows that the fastest urIPSCs were detected within 2 ms from TPU onset, implying that there is a fast mechanism coupling $Ca^{2+}$ entry to release - in line with earlier findings of tight coupling between $Ca^{2+}$ entry and GABA release (using EGTA, *Isaacson, 2001*), and of a crucial role for $Na_v$ channels (*Bywalez et al., 2015*; *Nunes and Kuner, 2018*) – as also indicated by our simulations (*Figure 5A*).

While ~30% of urIPSCs occurred within 10 ms post TPU onset, there was a substantial fraction with longer latencies in the range of 10–30 ms and a few with even larger delays (which might also correspond to false positives of our detection method). Again, since $V_m(t)_{SPINE}$ is unknown, we cannot determine to what extent these urIPSCs were actually asynchronous (if defined as release events that happen later than the fast coupling of HVA presynaptic $Ca^{2+}$ currents to the release machinery, for example *Kaeser and Regehr, 2014*). In any case, substantial asynchronous release from the GC spine on yet longer time scales (detected at up to 500 ms post TPU) was frequently observed, in line with earlier studies on recurrent inhibition that localized the origin of asynchronous signaling within GCs (*Schoppa et al., 1998*; *Chen et al., 2000*; *Isaacson, 2001*). This time course of asynchronous release also matches with the duration of physiological postsynaptic GC spine $Ca^{2+}$ transients (*Ona Jodar et al., 2020*). Thus, the microcircuit can operate across a wide range of latencies, which might contribute to glomerulus-specific GC global AP firing latencies (*Kapoor and Urban, 2006*), and also can generate combined synchronous and asynchronous output.

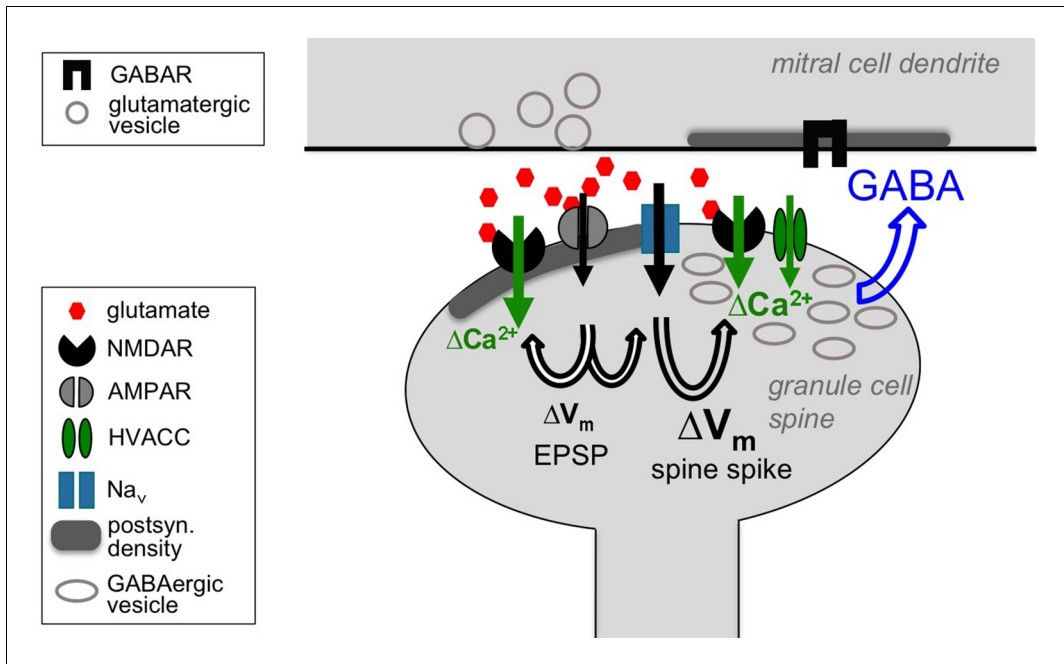

**Figure 7.** Cooperative release of GABA from the granule cell reciprocal spine. Depolarizing currents are indicated by black solid arrows and $Ca^{2+}$ entry by green solid arrows. Upon release of glutamate from the mitral cell dendrite, AMPARs get activated, leading to depolarization of the spine, then activation of both post- and presynaptic NMDARs and $Na_v$s. The latter in turn drive further depolarization, resulting in the spine spike, which both activates HVACCs and enhances $Ca^{2+}$ entry via NMDARs by additional relief of their $Mg^{2+}$ block. HVACC- and presynaptic NMDAR-mediated $Ca^{2+}$-currents cooperate to promote recurrent release of GABA onto the mitral cell dendrite.

## Na$_v$-mediated and NMDAR-mediated contributions to release

How is postsynaptic Ca$^{2+}$ entry coupled to release of GABA within the GC spine? Previously we had shown via occlusion experiments that Ca$^{2+}$ entry via NMDARs occurs independently from Ca$^{2+}$ entry mediated by the Na$_v$-HVACC pathway, since AMPAR-mediated depolarization on its own is strong enough to lift the Mg$^{2+}$ block, probably due to boosting by the high GC spine neck resistance (*Bywalez et al., 2015*). Therefore, we initially hypothesized that the Na$_v$-HVACC pathway would provide the sole trigger for fast release of GABA, as in classical release from axons (see also *Nunes and Kuner, 2018*), validating the notion of the GC spine as an independent mini-neuron that can generate output via its own local AP. Indeed, blockade of either Na$_v$ or HVACCs strongly reduced or abolished urIPSCs. However, in subsequent experiments probing NMDAR contribution we observed that urIPSCs were also massively reduced by blockade of NMDARs.

As a note of caution, activation of single GC spines via TPU might involve spurious activation of extrasynaptic NMDARs. We had observed that TPU resulted in a slightly larger NMDA-receptor-mediated component of the postsynaptic Ca$^{2+}$ signal than true synaptic activation via glomerular stimulation (~65% vs ~50% of ΔF/F *Bywalez et al., 2015*; *Egger et al., 2005*). Thus, at least part of the strong impact of NMDARs observed here might have been rooted in their over-activation. Therefore, we investigated recurrent inhibition elicited by single MC APs, and could demonstrate that NMDAR blockade alone (which does not prevent GC spine spike generation, *Bywalez et al., 2015*) also reduces recurrent inhibition. This effect was found to be mutually occlusive with the effect of GABA$_A$R blockade, arguing again in favor of an essential role of NMDARs for GABA release and against systematic overstimulation in the TPU experiments (see also Results, Materials and methods).

Another interesting aspect of the strong influence of NMDARs on GC output is that this property might serve to differentiate the MC↔GC microcircuit from the MC↔PV+ cell microcircuit, since PV+ cells feature Ca$^{2+}$-permeable AMPARs and a probably absent NMDAR component in response to MTC input (*Kato et al., 2013*). Thus, the urIPSC blockade by APV observed here further argues in favor of a preferential activation of the MC↔GC circuit by our experimental method.

In our earlier study of TPU-evoked Ca$^{2+}$ signals in GC spines there was no obvious influence of Na$_v$ activation on NMDAR-mediated Ca$^{2+}$ entry (*Bywalez et al., 2015*). At first, this finding might seem at variance with the simulations in *Figure 5* that demonstrate a boosting of early NMDAR-mediated Ca$^{2+}$ currents by the spine spike. However, the simulations also indicate that this extra contribution is not substantial in terms of added ΔF/F under our experimental conditions and therefore could not be detected (*Figure 5D*). In line with the fast NMDAR-mediated Ca$^{2+}$ spikelet, *Kampa et al., 2004* have shown that the earlier the postsynaptic membrane is depolarized after glutamate release, the more efficiently NMDARs will be activated. These observations are of general importance in the context of spike-timing dependent plasticity and electrical compartmentalization of spines (e.g. *Grunditz et al., 2008*; *Tønnesen and Nägerl, 2016*).

How exactly do NMDARs now effect release? The Na$_v$/NMDAR-mediated Ca$^{2+}$ spikelet results in substantial and fast coincident activation of NMDARs and HVACCs, and our ultrastructural evidence for presynaptic NMDARs implies that indeed these currents can feed into the same presynaptic microdomain (see Results). This overlap bonus at the presynaptic Ca$^{2+}$ sensor (as predicted by the simulations) could thus explain the observed cooperative signaling of HVACCs and NMDARs; further investigations are required to consolidate or replace this hypothesis.

## Direct involvement of presynaptic NMDARs in GABA release

Presynaptic actions of NMDARs at specific subsets of both glutamatergic and GABAergic synapses have recently received attention (reviewed *in Bouvier et al., 2015*; *Banerjee et al., 2016*). Presynaptic NMDARs have been shown to play a role in plasticity induction (e.g. *Duguid and Smart, 2004*) or in modulation of basal synaptic transmission, where effects are mostly observed upon repetitive transmitter release (e.g. *McGuinness et al., 2010*). In the cerebellum, presynaptic NMDARs are also involved in enhancing spontaneous release of GABA (*Glitsch and Marty, 1999*), with similar observations for extrasynaptic NMDA receptors on retinal A17 amacrine cells (*Veruki et al., 2019*). However, to our knowledge direct triggering of release via presynaptic NMDARs during unitary transmission has not been observed so far, adding another pathway to the already highly diverse signaling downstream of NMDARs.

The observed cooperation between Na$_v$/HVACCs and NMDARs relates our study back to the initial studies on dendrodendritic recurrent inhibition (DDI), when it was concluded by several groups that NMDARs can contribute directly to release from the reciprocal spine upon extended release of glutamate from the MC dendrite (see Introduction), and *Schoppa et al., 1998* already speculated that this presynaptic function might be related to either Ca$^{2+}$ entry or depolarization. However, the relative contribution of NMDARs has been under debate and it was demonstrated that under physiological conditions HVACC activation would be the dominant release trigger (*Isaacson, 2001*). While the standard DDI protocol, using 20–50 ms long depolarizations, would evoke recurrent inhibition also in the presence of TTX (possibly even enhanced, *Halabisky et al., 2000*), it was reported that recurrent inhibition in response to shorter steps (<5 ms) is substantially smaller than for the long step and reduced in TTX (*Schoppa et al., 1998*; *Halabisky et al., 2000*). In view of our above results, the standard DDI protocol is likely to recruit NMDAR-dependent pathways for triggering GABA release via prolonged release of glutamate and subsequent summation of EPSPs in GC spines, whereas short steps are more likely to trigger release via the GC spine spike. Thus, the cooperation of NMDARs and HVACCs reconciles these earlier findings.

## Reciprocal inhibition and spontaneous inhibitory activity

Spontaneous IPSCs are likely to predominantly originate from the more proximal lateral MC dendrites (*Arnson and Strowbridge, 2017*). The slightly larger mean amplitude of sIPSCs compared to the triggered urIPSCs observed here might be explained by perisomatic inhibitory contacts onto MCs, for example from type S GCs (*Naritsuka et al., 2009*). All three antagonists of urIPSC generation (TTX, CTX, APV) were found to also substantially reduce sIPSC frequency, which might imply that their effects on urIPSCs were due to a general reduction in excitability or network effects. However, none of the antagonists affected sIPSC amplitude or reduced sIPSC frequency in a manner correlated with their effects on the evoked urIPSCs, and similar effects on spontaneous activity were also reported elsewhere. Na$_v$ blockade was already shown to strongly reduce MC sIPSC frequencies but not amplitude (e.g. *Saghatelyan et al., 2005*; *Arnson and Strowbridge, 2017*), which is to be expected because of the blockade of both spontaneous MC firing and the GC spine spike. In our experiments the effect of HVACC blockade on the frequency of spontaneous events was less pronounced than that of Na$_v$s, probably because spontaneous MC firing is less affected.

A substantial effect of NMDAR blockade on sIPSC frequencies was also observed in other olfactory bulb studies (*Wellis and Kauer, 1993*; *Schmidt and Strowbridge, 2014*). Several factors might contribute: (1) the long-lasting depolarizations in MCs that are enhanced by NMDAR activation (*Carlson et al., 2000*), (2) the presence of MC NMDA autoreceptors (*Isaacson, 1999*; *Sassoè-Pognetto et al., 2003*), (3) the crucial role of NMDARs for reciprocal release from the GC spine reported here, thus any spontaneous release of glutamate from MCs is unlikely to trigger recurrent IPSCs in the presence of APV. (4) Finally, it is conceivable that the enhancement of spontaneous activity by the caged compound (*Figure 1—figure supplement 1B*) results in slightly elevated levels of ambient glutamate and thereby further augments the contribution of NMDARs to spontaneous activity via the three aforementioned mechanisms.

## Functional role of presynaptic NMDARs in GC spines: linking coactive glomerular columns?

While a presynaptic contribution of NMDARs to GABA release from GCs was already demonstrated earlier (see above), our findings imply that not only can NMDAR activation trigger GABA release, it is actually necessary, also in the presence of a Na$_v$-mediated spine spike. This observation is very intriguing for several reasons, that we will briefly discuss here:

1. Role of GC APs for release from the reciprocal spine.
   Notwithstanding the precise site of AP generation in GCs, any AP will propagate along the GC dendrite and into the reciprocal spines perfectly well (*Egger et al., 2003*; *Pressler and Strowbridge, 2019*). The ensuing spine depolarization V$_m$(t)$_{SPINE}$ will resemble that of the spine spike (see also *Aghvami et al., 2019*) which on its own, with NMDARs blocked, according to our findings above cannot trigger GABA release. Therefore, any type of non-local GC AP is rather unlikely to cause release of GABA by itself. This consequence explains the lack of functional evidence for GC → MC transmission in coupled MC → GC pairs, because APs elicited in the GC will not result in GABA release and thus no inhibition will be exerted on the

coupled MC (see Introduction).It should be noted that extracellular stimulation of the GC layer can elicit MC inhibition, even in the presence of APV (*e.g. Chen et al., 2000*, *Egger et al., 2003*, *Arevian et al., 2008*). Possible explanations might involve either a small remaining probability for GABA release from spiking GCs in the absence of glutamate or that this type of stimulation activates some as of yet unknown inhibitory input to MCs, for example via deep short-axon cells or axons of EPL interneurons (*Nagayama et al., 2014*; *Burton, 2017*).

2. Role of GC-mediated lateral inhibition between MCs.

It is known by now that broad, far-range lateral inhibition between MCs within the EPL is mediated by other interneurons than GCs (see Introduction). Thus, GCs are unlikely to dominate bulbar lateral inhibition in terms of magnitude, i.e. total charge transfer, a view that is also supported by other recent studies (*Fukunaga et al., 2014*; *Geramita and Urban, 2017*). What then is the function of GC-mediated lateral inhibition?

As a consequence of the requirement of presynaptic NMDAR activation, we predict that there will be no or at best very little GC-mediated lateral inhibition upon activation of a single glomerular column, even if some columnar GCs fire APs or dendritic $Ca^{2+}$ spikes (which is fairly likely, *Mueller and Egger, 2020*). We propose that GC-mediated lateral inhibition happens predominantly across coactive columns. Thus the discovered mechanism would allow GCs to perform lateral inhibition 'on demand', selectively on co-active MCs, providing directed, dynamically switched lateral inhibition in a sensory system with large numbers of receptor channels. GC-mediated lateral inhibition might therefore participate in synthesizing the olfactory percept at the level of the bulb from the individual active olfactory receptor channels, most likely via MC synchronization in the gamma band (e.g. *Laurent et al., 1996*; *Kashiwadani et al., 1999*; *Schoppa, 2006*; *Brea et al., 2009*; *Li et al., 2015*; *Peace et al., 2017*), while at the same time preventing unnecessary energy expenditure (i.e. if GABA release would happen from all reciprocal spines of an activated GC). More importantly, inactive columns will remain sensitive for new stimuli or changing components because they are not inhibited by active columns.

There is substantial prior evidence in the literature for both the activity-dependence and NMDAR-dependence of GC-mediated lateral inhibition (*Arevian et al., 2008*). NMDAR blockade is routinely used to dissect lateral inhibition in the glomerular layer from GC-mediated lateral inhibition (*Shao et al., 2012*; *Najac et al., 2015*; *Geramita and Urban, 2017*). Yet more importantly, in in vivo MC recordings there was a subset of MCs that responded exclusively with inhibition to odor activation, mediated via glomerular layer neurons but not via GCs (*Fukunaga et al., 2014*, their Figures 4 and 7). Since there was no odor-evoked activity in these MCs, our results imply that the presynaptic GC NMDARs could not be activated, preventing GABA release from GCs.

As to behavioral evidence for our hypothesis, both GC NMDARs and $Na_v$s are known to play a role specifically in discriminations between binary odorant mixtures in mice, since in the same behavioral paradigm GC-specific knock-down (in about ~50% of GCs) of the NMDAR-subunit GluN1 resulted in an increase in discrimination time of on average ~60 ms, and knock-down of $Na_v$s of ~85 ms vs control, while discriminations between pure odors were not affected (*Abraham et al., 2010*; *Nunes and Kuner, 2018*). These similar effects are in line with our findings, since both modifications should prevent both recurrent and lateral inhibition. However, the relative impact of recurrent inhibition vs lateral inhibition is not known.

## Conclusion: Hypothesis on the purpose of the reciprocal spine

In summary, our findings suggest that reciprocal spines allow the GC to selectively interact with coactive glomerular columns. Thus, the reciprocal spine might be a circuit motif that specifically enables efficient binding of dispersed olfactory representations, a coding task that this sensory modality needs to perform due to its large number of receptors. In other words, reciprocal spines with their unique synaptic arrangement and functionality, including spine spikes and presynaptic NMDARs, are likely to represent a special adaptation to the demands of sensory processing in the bulb. Of course, at this point this is a hypothesis that awaits further exploration via network simulations and in vivo approaches. Another intriguing yet more speculative potential function of target-activity-dependent output is that concurrent preactivation from glutamatergic cortical inputs onto reciprocal spines (ultrastructure in *Price and Powell, 1970*) can also enhance lateral inhibition in addition to providing feed-forward inhibition, potentially allowing for a top-down projection of olfactory templates as has been proposed by *Zelano et al., 2011*.

## Materials and methods

### Animal handling, slice preparation and electrophysiology

Animals used in our experiments were juvenile Wistar or VGAT-Venus transgenic rats (VGAT-Venus/w-Tg(SLc32a1-YFP*)1Yyan) of either sex (P11 – P19). VGAT-Venus transgenic rats are based on mouse BAC transgenic lines. They were generated by Drs. Y. Yanagawa, M. Hirabayashi and Y. Kawaguchi at the National Institute for Physiological Sciences, Okazaki, Japan, using pCS2-Venus provided by Dr. A. Miyawaki (**Uematsu et al., 2008**, RRID:RGD_2314361). In this rat line, fluorescent Venus protein is preferentially expressed in cells carrying the vesicular GABA transporter (VGAT), that is GABAergic neurons: the localization of Venus-labeled cells across OB layers was found to be similar to that of GABA-positive cells; direct colocalization in the cortex yielded an overlap of 97% (**Uematsu et al., 2008**).

Sagittal olfactory bulb brain slices (thickness 300 µm) were prepared in ACSF (composition see below) following procedures in accordance with the rules laid down by the EC Council Directive (86/89/ECC) and German animal welfare legislation. Slices were incubated in a water bath at 33°C for 30 min and then kept at room temperature (22°C) until recordings were performed.

Olfactory bulb MCs were visualized by gradient contrast and recorded from in whole-cell voltage-clamp mode (at −70 mV or +10 mV) or current clamp mode. Recordings were made with an EPC-10 amplifier and Patchmaster v2.60 software (both HEKA Elektronik, Lambrecht/Pfalz, Germany). Experiments were performed at room temperature (22°C). Patch pipette resistance ranged from 5 to 6 MΩ. MCs clamped at −70 mV were filled with intracellular solution containing the following substances (in mM): (1) tip solution: 130 K-Methylsulfate, 10 HEPES, 4 MgCl2, 2 Ascorbic acid, 10 Phosphocreatine-di-tris-salt, 2.5 Na2ATP, 0.4 NaGTP (2) backfilling solution: 110 Cs-Chloride, 10 HEPES, 10 TEA, 4MgCl2, 2 Ascorbic acid, 10 5-N-(2,6-dimethylphenylcarbamoylmethyl) triethylammonium bromide (QX-314, Sigma), 0.2 EGTA, 10 Phosphocreatine, 2.5 Na2ATP, 0.4 NaGTP, 0.05 Alexa 594 ($Ca^{2+}$ indicator, Thermofisher Scientific, Waltham. Massachusetts, US), at pH 7.3. MCs clamped at + 10 mV contained internal solution composed of: 125 Cs-methanesulfonate, 1 NaCl, 0.5 EGTA, 10 HEPES, 3 MgATP, 0.3 NaGTP, 10 Phosphocreatine-di-Tris-salt, 10 QX-314, 0.05 Alexa 594, at pH 7.3.

For current clamp experiments in either MCs or GCs the internal solution contained: 130 K-Methylsulfate, 10 HEPES, 4 MgCl2, 2 Ascorbic acid, 10 Phosphocreatine-di-Tris-salt, 2.5 Na2ATP, 0.4 NaGTP. Single APs were evoked by somatic current injection (3 ms, 1 nA) and in MCs 5 APs were elicited for every recording condition. MCs with leak currents > 200 pA and GCs with leak currents > 50 pA (at $V_{hold}$ = - 70 mV) were discarded. For MCs, we chose a hyperpolarized holding potential in order to reduce the activation of NMDA autoreceptors on MC lateral dendrites.

The extracellular ACSF was bubbled with carbogen and contained (in mM): 125 NaCl, 26 $NaHCO_3$, 1.25 $NaH_2PO_4$, 20 glucose, 2.5 KCl, 1 $MgCl_2$, and 2 $CaCl_2$. The following pharmacological agents were bath-applied in some experiments: bicuculline (BCC, 50 µM, Sigma-Aldrich), ω-conotoxin MVIIC (CTX, 1 µM, Alomone, Jerusalem, Israel), TTX (500 nM, Alomone), D-APV (25 µM, Tocris), gabazine (GBZ, 20 µM, Tocris). In pharmacological experiments we waited for 10 min after wash-in of the drugs TTX, APV resp. CTX. In CTX experiments 1 mg/ml cyctochrome C was added to the ACSF. TTX voltage-clamp experiments were conducted at clamping potentials of −70 mV (n = 5 MCs) or + 10 mV (n = 7 MCs), whereas all CTX and APV voltage-clamp experiments were conducted at + 10 mV.

### Combined two-photon imaging and uncaging

Imaging and uncaging were performed on a Femto-2D-uncage microscope (Femtonics, Budapest, Hungary). Two tunable, Verdi-pumped Ti:Sa lasers (Chameleon Ultra I and II respectively, Coherent, Santa Clara, CA, USA) were used in parallel. The first laser was set either to 900 nm for simultaneous excitation of YFP and Alexa 594 in one channel for visualization of spines and the MC for urIPSC recordings (or the GC for uEPSP recordings), and the second laser was set to 750 nm for uncaging of caged glutamate. The two laser lines were directly coupled into the pathway of the microscope with a polarization cube (PBS102, Thorlabs Inc, Newton, NJ, USA) and two motorized mirrors. As caged compound we used DNI-caged glutamate (DNI; Femtonics). DNI was used in 1 mM concentration in a closed perfusion circuit with a total volume of 12 ml. Caged compounds were washed in

for at least 10 min before starting measurements. The uncaging laser was switched using an electro-optical modulator (Pockels cell model 350–80, Conoptics, Danbury, CT, USA). The emitted fluorescence was split into a red and a green channel with a dichroic mirror.

## Triggering of MC reciprocal IPSCs

The region of interest on a proximal lateral MC dendrite was moved to the center of the scanning field. In some initial experiments, four uncaging spots were placed in close apposition along the lateral MC dendrite in case of 'blind' uncaging (n = 8 out of total number of successful experiments in n = 44 MCs). In most experiments, a single uncaging spot was positioned near the region of interest, using YFP fluorescence of GABAergic cells in VGAT-Venus rats as an optical guide to identify potential synaptic contacts between MC lateral dendrites and GC spines.

The parameter settings for two-photon uncaging were as established previously on the same experimental rig for precise mimicking of unitary synaptic $Ca^{2+}$ transients in GC spines (*Egger et al., 2005*; *Bywalez et al., 2015*). These parameter settings were routinely verified in parallel test experiments, where we imaged $Ca^{2+}$ transients in GC spines following TPU (as in *Bywalez et al., 2015*). The uncaging site was usually chosen within the top 10–30 µm below the slice surface (since otherwise the visibility of spines in the Venus-VGAT rats was compromised) and the uncaging power was adjusted to the depth of the uncaging site, corresponding to a laser power of approximately 15 mW at the uncaging site (*Sobczyk et al., 2005*). The uncaging beam was positioned at ~0.2–0.5 µm distance from the MC dendrite/GC spine head. The uncaging pulse duration was 1 ms. The scanning position was readjusted if necessary before each measurement to account for drift. The microscope was equipped with a 60x Nikon Fluor water immersion objective (NA 1.0; Nikon Instruments, Tokyo, Japan). The microscope was controlled by MES v.5.3190 software (Femtonics). To prevent overstimulation, the uncaging laser power was **not** increased if there was no detectable response to the preset laser power (based on depth). While we cannot exclude overstimulation per se, we would like to argue that systematic overstimulation is unlikely to have occurred based on the following observations (see also Results):

- size of urIPSCs as compared to spontaneous activity (*Figure 1E*)
- low reciprocal release probability from the spine (*Figure 1E*)
- no extended asynchronous release compared to the classical experiments on recurrent inhibition (mean total duration of TPU-evoked recurrent inhibition ~200 ms, *Figure 1—figure supplement 1G*, vs a mean half duration ~500 ms in *Schoppa et al., 1998*, *Isaacson, 2001*)
- no evidence for toxicity in our stability experiments (*Figure 1F*)
- main finding of dependency of recurrent inhibition on NMDAR activation confirmed by method that does not involve uncaging

## Uncaging stability

To test for the stability of urIPSCs in MCs, uncaging at a dendritic region of interest was performed. If repeated uncaging led to the apparent occurrence of urIPSCs within the same time window (see below for analysis details), the stability measurement was continued by either uncaging 30 times in total with a frequency of 0.033 Hz (every 30 s) or five times in a row with 0.033 Hz followed by a 10 min break (to mimic the time for wash-in of pharmacological compounds) and another round of uncaging (5x, 0.033 Hz). urIPSC amplitudes were taken from averages of the first 5–10 and the last 5–10 uncaging sweeps and statistically compared with each other.

## Uncaging onto nearby GC spines

For the control experiments where we tested whether adjacent GC spines could sense the glutamate uncaged at an individual spine (*Figure 1—figure supplement 1C,D,E*), uncaging was performed as described above and in *Bywalez et al., 2015* near a spine head of a GC filled with 100 µM OGB-1 and 50 µM Alexa594 in whole-cell current clamp mode. $Ca^{2+}$ imaging was performed using a line scan through the spine head activated by TPU, the adjacent spine head and the dendritic shaft in between.

### Electrophysiology: Data analysis and statistics

Electrophysiological data were analyzed with custom written macros in Igor pro (Wavemetrics, Lake Oswego, OR, USA). Additional sIPSC and urIPSC analysis was performed using the MiniAnalysis program (Synaptosoft, Decature, GA, USA) and Origin (Northampton, MA, USA).

#### Detection of urIPSCs

Due to the high spontaneous activity, in order to test for the presence of a signal we performed first an area and event analysis of IPSC traces (see below and *Figure 1D*); if a signal was detected based on these analyses, we went on to search for individual triggered urIPSCs by visual inspection of an overlay of the recorded traces. Individual IPSCs were considered as uncaging-evoked when they repetitively occurred within the same time window (width 3 ± 2 ms, n = 35 MCs) after uncaging and had similar kinetics (indicating a similar location of the respective input on the MC dendrite). Signal types ranged from single urIPSC events to barrages of urIPSCs lasting tens to hundreds of ms. The release probability $P_{r\_GABA}$ was estimated based on 5–30 TPU samplings with a mean of 7.5 ± 1.7 stimulations (n = 44 MCs).

#### Area analysis

The area was measured in individual traces as the integrated activity above baseline for a 500 ms pre-uncaging baseline window and for a 500 ms post-uncaging window, in order to screen for the presence of a signal (*Figure 1D*). The 500 ms extent of the time windows was validated by our measurements of averaged barrage duration (see *Figure 1—figure supplement 1G*).

Delta (Δ) area values were calculated by subtracting the area of the 500 ms pre-uncaging baseline window ('pre') from the 500 ms post-uncaging window ('post'), in order to isolate the amount of uncaging-evoked inhibitory activity from spontaneous activity. If this procedure was applied to averaged traces and the result was negative, the Δ_area value was set to zero (i.e. no uncaging-evoked activity). While this procedure might still generate false positives due to spontaneous bursts of activity in the post-uncaging window, it also prevents a spurious cancelling of activity (false negative) that otherwise might happen upon averaging Δ_area across an entire set of experiments. Δ_area values for pharmacological conditions were normalized to control Δ_area in order to assess the net effect of drugs on uncaging-evoked inhibitory activity (*Figures 2D*, *3D* and *4D*).

#### Event analysis

Within the individual recorded traces, the peak time points of individual IPSCs were analyzed. Peak search parameters in MiniAnalysis were adjusted in order to detect potentially all IPSCs within a trace. For detailed spontaneous IPSC amplitude analysis, IPSCs were sorted manually after the automated peak search and discarded if the amplitude exceeded less than 5 pA and/or the IPSC onset was not be detected properly. Event counts were averaged for the 500 ms pre-uncaging and the 500 ms post-uncaging windows, respectively.

#### Evaluation of effects of pharmacological agents

For determining drug effects, the averaged urIPSC amplitudes were scaled down by the ratio of number of responses to total number of trials, both in control and drug condition, in order to account also for changes in release probability. If no single responses/urIPSCs could be detected anymore in the presence of TTX, CTX, or APV according to the criteria described above, we measured the mean amplitude of $I_m$ above baseline in the averaged response at the time point of the maximal response amplitude in control condition. If this value was below 0, the response size was set to 0 pA. If the value was larger than 0, we interpreted it as average drug response amplitude including failures and thus did not scale it. This conservative method prevents false negatives due to lacking sensitivity in individual trials in the presence of the drug.

#### Detection of spontaneous activity

Spontaneous IPSCs were recorded prior to wash-in of DNI, in the presence of DNI and in the presence of each pharmacological compound. For each condition, data were analyzed for a total duration of 20 s of recordings.

### Analysis of afterhyperpolarizations

All stable MC AP recordings within either baseline or drug condition(s) were averaged (n = 5 recordings each). If between the conditions the holding membrane potential changed by more than 0.3 mV, or the time course (onset of upstroke relative to onset of step depolarization, width) and/or the amplitude of the AP changed by more than 15% from their baseline values, the experiment was discarded. If single individual recordings showed such variations, they were not included in the average. For each average, the AHP amplitude was measured as the maximal negative deflection of the membrane potential from the resting membrane potential.

## Simulations

The simulations are based on a published compartmental model in NEURON (*Aghvami et al., 2019*, ModelDB entry 244687). This model uses the 5-state gating model for NMDA receptors (*Destexhe et al., 1998*), while the HVACC model is adopted from *Hemond et al., 2008*, with adjustments based on own $Ca^{2+}$ current recordings (see *Aghvami et al., 2019*). For the simulations shown in *Figure 5*, there was no exogenous $Ca^{2+}$ buffer included except for panel D where fluorescence transients were simulated in the presence of 100 µM of the $Ca^{2+}$-sensitive dye OGB-1 in order to emulate the experimental situation from *Bywalez et al., 2015*.

As a readout measure for the temporal overlap between $I_{Ca\_NMDAR}$ and $I_{Ca\_HVACC}$ we first determined the FWHM (full width half maximum) for each current and then the interval within which the two FWHMs overlapped. At this point, we focussed on temporal relationships and did not account for current amplitudes or integrals, since $Ca^{2+}$ concentration changes within nanodomains cannot be properly simulated in NEURON and thus spine $Ca^{2+}$ current amplitudes are of limited meaning. We tested for the robustness of this measure against variation of those model parameters that are crucially involved in the generation of the spine spike, that is the resistance of the spine neck ($R_{neck}$), the conductance of AMPA receptors ($g_{AMPA}$) and the voltage-gated sodium channel conductance (varied in proportion to the potassium channel conductance, $g_{Na/K}$). Their nominal values in the GC model are 1.7 GΩ, 2000 pS and 0.5 S/cm$^2$ respectively. Each parameter was varied between up to 200% of the nominal value, and down until no spine spike was generated any more.

## Immunogold labeling and electron microscopy

Immunogold labeling was performed on cryosubstituted rat olfactory bulbs (n = 4 animals, 3 months old), that had been used previously; for further details on the fixation, embedding and immunogold labeling procedure see *Sassoè-Pognetto and Ottersen, 2000*; *Sassoè-Pognetto et al., 2003*.

To maximize detection of the GluN1 subunit, we used a combination of two rabbit antisera as described in *Sassoè-Pognetto et al., 2003*. One antiserum (kindly donated by Anne Stephenson) binds an extracellular domain (amino acid residues 17–35) common to all splice variants of the GluN1 subunit (*Chazot et al., 1995*; *Racca et al., 2000*). The other antiserum was raised against a C-terminal domain and recognizes four splice variants (Chemicon, Temecula, CA; cat. no. AB1516).

For GluN2, we used a affinity-purified rabbit antibody raised against a synthetic peptide corresponding to the C-terminus of the GluN2A subunit conjugated to BSA (Chemicon, cat. no. AB1548). According to the manufacturer, this antibody recognizes the GluN2A and GluN2B subunits in Western blot analysis of transfected cells.

## Ultrastructure data analysis and statistics

Grid squares were analyzed systematically for the presence of synaptic profiles (symmetric and/or asymmetric) between GC spines and MC dendrites (Figure 6; see also Figure 1A in *Sassoè-Pognetto and Ottersen, 2000*). Synaptic profiles were then photographed at high magnification (75.000–120.000x) with a side-mounted CCD camera (Mega View III, Olympus Soft Imaging System). The plasma membrane of GC spines, when clearly visible, was classified as either belonging to an asymmetric synaptic profile, a symmetric profile, or a non-synaptic segment (see *Figure 6A2* for examples). The length of segments was measured along the spine membrane curvature (using ImageJ 1.52 analysis software) and the number of immunogold particles within a distance of ≤30 nm from the GC spine membrane was counted for the individual segments. The lengths of non-synaptic segments were on average longer than those of synaptic segments, which argues against an undersampling of gold particle densities in non-synaptic membranes compared to synaptic membranes

and thus a false positive difference between the density distribution in non-synaptic membranes and symmetric profiles (GluN1: mean non-synaptic segment length: 610 ± 400 nm, n = 138; symmetric synaptic profiles: 270 ± 120 nm, n = 120; asymmetric synaptic profiles: 260 ± 110 nm, n = 111; similar results for GluN2, not shown). When both symmetric and asymmetric synaptic profiles were visible in the same individual spine, the distance of such reciprocal contacts was also measured along the curvature of the GC spine membrane.

The distribution of labeling along the axis perpendicular to the GC spine membrane (radial axis) was determined by examining micrographs of transversely cut synaptic profiles, with well defined presynaptic and postsynaptic membranes. Here all particles at distances up to 40 nm were counted to prevent a possible bias.

## Statistical tests

All replicates were biological, that is physiological experiments were repeated in different cells and ultrastructural analyses were conducted in different samples. Since sample sizes were too small to test for the normal distribution of data and the main type of experiment was novel (i.e. two-photon uncaging of glutamate on spines and detection of reciprocal IPSCs), all electrophysiological data were analyzed with non-parametric paired (*Wilcoxon matched* pairs, e.g. for comparing data before and after wash-in of a drug) or unpaired (*Mann-Whitney-U,* for comparing effects of different drugs) tests and expressed as mean ± SD. Based on the properties of these tests and our long-term experience with non-parametric testing (since *Egger et al., 2003*), we used a sample size of at least n = 7 for control type experiments (e.g., stability of urIPSC recordings) and n = 10 for core experiments (effects of $Na_v$ and NMDAR blockade). For the main findings, effect sizes (Cohen's d) were calculated using G*Power (3.1, https://www.psychologie.hhu.de/arbeitsgruppen/allgemeine-psychologie-und-arbeitspsychologie/gpower.html). The density distributions of immunogold particles were compared with the Kolmogorov-Smirnov test, which also does not make prior assumptions on the nature of data distributions (https://www.wessa.net/rwasp_Reddy-Moores%20K-S%20Test.wasp).

All Figures: Significance (all non-parametric tests): n.s. not significant, *p<0.05, **p<0.01, ***p<0.001; mean data ± S.D.

## Acknowledgements

We thank Anne Pietryga-Krieger for expert technical assistance, Marius Stephan and Philipp Seidel for assistance with IPSC analysis, Gagik Yeghiazaryan and Anna Bartalis for help with MC current clamp recordings, Imre Vida for facilitating access to VGAT-Venus rats, Sigrun Korsching and Yoshiyuki Kubota for advice on statistics and Ursula Koch for advice on pharmacology and discussions on lateral inhibition. This work was funded mainly by the German Federal Ministry for Education and Research (BMBF, 01GQ1104/01GQ1502), with additional equipment funding by LMU-GSN, DFG-SFB 870 and funding for staff from GIF (1479–418.13; all to VE), and by ERC 682426, GINOP_2.1.1-15-2016-00979, VKSZ_14-1-2015-0155, 712821-NEURAM to BR.

## Additional information

### Competing interests

Balázs Rózsa: BR is a founder of Femtonics Kft and a member of its scientific advisory board. No other competing interests exist. The other authors declare that no competing interests exist.

### Funding

| Funder | Grant reference number | Author |
| --- | --- | --- |
| Bundesministerium für Bildung und Forschung | 01GQ1104/01GQ1502 | Veronica Egger |
| German-Israeli Foundation for Scientific Research and Development | 1479-418.13 | Veronica Egger |
| Graduate School of Systemic | DFG SFB 870 | Veronica Egger |

| | | |
|---|---|---|
| Neurosciences GSN-LMU | | |
| European Research Council | 682426 | Balázs Rózsa |
| European Research Council | GINOP_2.1.1-15-2016-00979 | Balázs Rózsa |
| European Research Council | VKSZ_14-1-2015-0155 | Balázs Rózsa |
| European Research Council | 712821-NEURAM | Balázs Rózsa |

The funders had no role in study design, data collection and interpretation, or the decision to submit the work for publication.

### Author contributions

Vanessa Lage-Rupprecht, Conceptualization, Data curation, Formal analysis, Investigation, Visualization, Methodology, Project administration, Writing - review and editing; Li Zhou, Investigation; Gaia Bianchini, Investigation, Writing - review and editing; S Sara Aghvami, Investigation, Methodology, Writing - review and editing; Max Mueller, Investigation, Visualization, Methodology; Balázs Rózsa, Resources, Writing - review and editing; Marco Sassoè-Pognetto, Supervision, Investigation, Writing - review and editing; Veronica Egger, Conceptualization, Data curation, Formal analysis, Supervision, Funding acquisition, Investigation, Visualization, Methodology, Writing - original draft, Project administration, Writing - review and editing

### Author ORCIDs

Veronica Egger https://orcid.org/0000-0002-5869-8523

### Ethics

Animal experimentation: All experimental procedures were performed in accordance with the rules laid down by the EC Council Directive (86/89/ECC) and German animal welfare legislation. According to this legislation (§4 Absatz 3 TierSchG), the preparation of acute brain slices for in vitro experiments by certified personnel (which applies to all experimenters in this study) is monitored by the institutional veterinarian of Regensburg University and does not require approval by an ethics committee. Rats (postnatal day 11-21, Wistar of either sex) were deeply anaesthetized with isoflurane and decapitated. Sagittal olfactory bulb slices were prepared.

### Decision letter and Author response

Decision letter https://doi.org/10.7554/eLife.63737.sa1
Author response https://doi.org/10.7554/eLife.63737.sa2

# Additional files

### Supplementary files

• Transparent reporting form

### Data availability

Figure source data deposited in Dryad Digital Repository (https://doi.org/10.5061/dryad.t4b8gtj04).

The following dataset was generated:

| Author(s) | Year | Dataset title | Dataset URL | Database and Identifier |
|---|---|---|---|---|
| Lage-Rupprecht V, Zhou L, Bianchini G, Aghvami SS, Mueller M, Rózsa B, Sassoé-Pognetto M, Egger V | 2020 | Source data for Fig. 1-6 and Fig. S1, S2,S3,S4 | http://dx.doi.org/10.5061/dryad.t4b8gtj04 | Dryad Digital Repository, 10.5061/dryad.t4b8gtj04 |

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
