## [Decision Letter]

**Acceptance summary:**

The authors have addressed the remaining concerns by revising the manuscript. This study examined the neural mechanism underlying recurrent inhibitions between mitral and granule cells in the rat olfactory bulb. The authors provide evidence supporting local interaction, likely involving a single granule cell spine, and contributions of various mechanisms such as NMDA receptors, voltage-gated sodium channels and high voltage-activated calcium channels in this process. They also provide simulation results and electron microscopic evidence supporting local interactions.

**Decision letter after peer review:**

[Editors’ note: the authors submitted for reconsideration following the decision after peer review. What follows is the decision letter after the first round of review.]

Thank you for submitting your work entitled "Presynaptic NMDARs cooperate with local spikes toward GABA release from the reciprocal olfactory bulb granule cell spine" for consideration by *eLife*. Your article has been reviewed by two peer reviewers, and the evaluation has been overseen by a Reviewing Editor and a Senior Editor. The reviewers have opted to remain anonymous.

Our decision has been reached after consultation between the reviewers. Based on these discussions and the individual reviews below, we regret to inform you that your work will not be considered further for publication in *eLife*.

Summary

Specifically, the authors examined the mechanism underlying reciprocal dendro-dendritic interactions between mitral cells and granule cells of the olfactory bulb in rats. Minimal stimulation with two-photon glutamate uncaging was used to stimulate a single spine of a granule cell while inhibitory postsynaptic currents (IPSCs) were monitored using whole-cell, voltage-clamp recording from a mitral cell. The authors show that recurrent IPSCs were greatly reduced by pharmacological inhibition of voltage-gated Na channels, high-voltage-activated Ca channels or NMDA receptors. The authors also provide electron microscopic (EM) data that shows that NMDA receptors are indeed located close to the release sites in granule cell spines. The authors proposed that NMDA receptor-mediated calcium influx facilitates synapse-specific lateral inhibition, and speculate its role in sensory processing.

The reviewers appreciated the high quality of the experiments and the data. Both reviewers found the proposed ideas very interesting. The EM data are also very intriguing. However, the reviewers had substantive concerns on the validity of the authors' interpretations of the experiment and the data. First, they were concerned that the manuscript does not provide compelling evidence that only a single spine in a granule cell was activated by glutamate uncaging even though many of the authors' conclusions depend on it. Second, all of the pharmacological manipulations caused a significant decrease in evoked as well as spontaneous IPSCs. It is therefore unclear whether the observed pharmacological effects were as interpreted or due to a mechanism that does not require an evoked event (e.g. a general reduction in excitability). Both reviewers also pointed out that the Discussion section includes speculative ideas that are not directly related to experimental results.

During discussion, one reviewer was more positive than another reviewer regarding the overall impact of the study. In any case, both reviewers agreed that the first two concerns mentioned above are substantive, and fully addressing these concerns likely requires additional experiments that will take more than 2-3 months (e.g. measuring the spread of excitation resulting from the authors' uncaging protocol). Based on these considerations, we have decided that we cannot proceed further with the current manuscript. If the authors can fully address these major concerns, however, we would be willing to reconsider a revised manuscript.

Reviewer #1:

Lage-Rupprecht and colleagues describe biophysical mechanisms of MC-GC recurrent inhibition using minimal stimulation of GC spines. They use a local two-photon uncaging of glutamate to evoke IPSCs that are smaller than spontaneous events. Surprisingly for a minimal stimulation of this magnitude, they observe that voltage gated Na channels, high voltage activated Ca channels as well as NMDA receptors are all necessary to evoke a recurrent IPSC. It should be noted that unlike previous studies in the field, the authors demonstrate the NMDA requirement under a physiological condition. Using an ultrastructural evidence, they demonstrate that NMDA receptors are strategically located i.e., in proximity to the release sites on the GC spine, well within the 100 nm boundary that theoretical studies suggest is crucial if the calcium influx lead directly to vesicle release. Compartmentalised Ca influx via a local NMDA activation would be an intriguing and ideal mechanism to implement specific yet flexible connectivity between MCs and GCs.

The data are of high quality, and the insights on the NMDA receptor involvement is new and very important. There is some confusion as to whether or not this is a study about a single-spine stimulation, while the majority of the discussion is spent on co-operative activation by several MCs. While it is not necessary that the data deals with a single-spine stimulation strictly, a manuscript may improve by clarifying this point. These are elaborated below in interrelated points.

1) Regarding the minimal stimulation used, the intention and interpretation seems to be that the two-photon uncasing leads to a single-spine stimulation. It is not clear that this is demonstrated. For example, in Figure 1X, the amplitude of IPSC shows some fluctuations. Does a histogram show a quantal nature or at least is the fluctuation consistent with a single contact?

2) According to a recent BioRxiv preprint (https://doi.org/10.1101/2020.01.10.901397 ), where some of the current authors describe Ca^2+^ dynamics in GC dendrites following a simultaneous uncaging of glutamate with incremental number of stimulated sites, demonstrating that Nav, HVACC, and NMDA receptors are recruited when several GC spines are stimulated, but not 1 spine. This may suggest the conditions used in the current manuscript may activate several spines. This may occur, for example, if more than 1 dendro-dendritic contacts exist between each GC-recorded MC pair, and clamping the recorded MC voltage at +10 mV (lower than the reversal potential of Ca^2+^ and high enough to activate the channel) could lead to some glutamate release on other parts of the GC dendrite. Have the authors tried to record at a more hyperpolarized voltage (maybe with a different internal solution to enhance the GABAergic conductance)?

3) Alternatively, it is possible that the apparent discrepancy above (between this and the previous studies) simply reflects a difference in the anatomical scale of analysis. That is, it is possible that the active conductance was already present with 1 stimulation site in the GC study, but highly localized within the spine head volume. So, an analysis of the spatial distribution of evoked calcium signal in GC spines may resolve this issue (if the data in the BioRxiv paper can be analyzed this way).

4) If indeed it turns out that the minimal stimulation involved a recruitment of several spines, it would be interesting to see if, in the Ca signals in the GC dendrite, the NMDA-receptor mediated calcium entry is confined to the activated spine (e.g., analysis of calcium signals in stimulated spine vs. non-stimulated spine next to it). This may support their idea of localized release during a more global event, such as during the co-operative activation by multiple MCs, as depicted in the schematics and elaborated in the Discussion.

5) Overall, the Discussion is quite lengthy and somewhat removed from the data currently presented. Please edit accordingly.

Reviewer #2:

This manuscript uses glutamate uncaging, whole cell recordings, and pharmacology to dissect the role of different currents in reciprocal inhibition at dendro-dendritic MC-GC synapses. Specifically, the authors record whole cell currents from MCs while performing glutamate uncaging onto nearby GCs and observe both spontaneous and evoked IPSCs in MC held at a depolarized potential. They then apply antagonists of GABA receptors, Nav channels, HVACCs, and NMDARs and show that in all cases both spontaneous IPSCs and evoked IPSCs are reduced. The results are interpreted as support for the "mini-neuron" hypothesis, in which single GC spines act as neurons with spike generation leading to HVACC activation and release. Further, the contribution of NMDARs to this process is interpreted as evidence for "on-demand" lateral inhibition between co-active columns.

The experiments shown appear to have been done carefully and the ideas proposed are interesting. However, I do think some of the results are over-interpreted here, given the strong suppression of spontaneous IPSCs observed with all pharmacological manipulations. The interesting ideas put forward in the lengthy Discussion might be effectively paired with a computational model to make an effective case in a manuscript of their own.

Major comments:

1) Overall the manuscript is written assuming a close familiarity with the prior literature on this circuit and synapses, and in particular with a previous manuscript from this lab. References to "the previous manuscript" are made frequently in the text without an exposition of what is shown there. The manuscript should be written as a stand-alone document that provides the necessary background for a reader to understand and interpret the findings as is.

2) All of the pharmacological manipulations shown decrease spontaneous IPSCs as well as IPSCs evoked by uncaging. Therefore, it is not clear of the mechanisms are specific to spine-evoked release or simply represent an overall decrease in excitability of the GC or network. At the very least evoked IPSC amplitude should be compared to baseline both before and after application of drugs.

3) The spatial extent of excitation produced by glutamate uncaging on GCs is not measured experimentally. While this excitation is assumed to be confined to a single spine this is never shown and seems to be critical to the interpretation. Voltage or Ca imaging from GCs during uncaging would be important to show this.

4) The Discussion is very long and puts forward a number of hypotheses that are significant extrapolations from the presented data. The ideas put forward here are quite interesting but I think would be more compelling if supported by a computational model and could easily constitute a manuscript of their own coupled with the extensive literature review that supports the model proposed.

[Editors’ note: further revisions were suggested prior to acceptance, as described below.]

Thank you for submitting your article "Presynaptic NMDARs cooperate with local spikes toward GABA release from the reciprocal olfactory bulb granule cell spine" for consideration by *eLife*. Your article has been reviewed by two peer reviewers, and the evaluation has been overseen by a Reviewing Editor and Gary Westbrook as the Senior Editor. The reviewers have opted to remain anonymous.

The reviewers have discussed the reviews with one another and the Reviewing Editor has drafted this decision to help you prepare a revised submission.

Summary

The revised manuscript was reviewed by the previous reviewer 1 and a new reviewer. They agreed that the revision addressed the previous concerns raised by the two reviewers. Reviewer 2 raised a new concerns and suggestions. We would like to ask you to revise the manuscript to address these remaining issues before making a final decision.

Reviewer #1:

In this manuscript, Lage-Rupprecht and colleagues describe mechanisms of MC-GC recurrent inhibition using minimal stimulations of GC spines. They use a local two-photon uncaging of glutamate to evoke IPSCs that are smaller than spontaneous events. Surprisingly for a minimal stimulation of this magnitude, they observe that voltage-gated Na channels, high voltage activated Ca channels as well as NMDA receptors are all necessary to evoke a recurrent IPSC. Using an ultrastructural evidence, they demonstrate that NMDA receptors are strategically located i.e., in proximity to the release sites on the GC spine, well within the 100 nm boundary that theoretical studies suggest is crucial if the calcium influx lead directly to vesicle release. A local activation of NMDA would be an intriguing and ideal mechanism to implement specific yet flexible connectivity between MCs and GCs. Their sophisticated experimental approach allowed a demonstration of such specific connectivity, involving a single GC spine.

In this revised version, the authors provide an experimental evidence (in Figure 2—figure supplement 1) that the calcium influx in the GC spine is local. Upon minimal two-photon stimulations, calcium signals are observed only in the stimulated spines, but not in neighboring spines or nearby dendritic segments. This strongly supports their model that a GABA-release from a GC spine relies on a glutamatergic input to that very spine, which leads to the NMDA-mediated Ca influx, thus demonstrating that GC-mediated inhibition can be synapse-specific. Their observation here also explains a variety of phenomena previously reported, such as an apparent lack of functional reciprocal connectivity when a GC depolarization on its own is used to probe GC->MC connectivity.

The discussion is now appropriate for the results presented.

I therefore conclude that the authors have addressed all of my previous concerns.

Reviewer #2:

The authors performed a series of high-quality experiments that make a strong argument for the mini-neuron model, with an interesting interpretation of these data that suggests that granule-cell-mediated lateral inhibition is specific to coactive glomerular columns.

Regarding the first major concern that they may be activating multiple spines, they now provide additional data that show only a small fraction of responses with blind uncaging, which is nice. However, more compellingly, when they do calcium imaging on two neighboring spines, they only see transients in the stimulated spine and not in its neighbor. Together, these data strengthen their argument that they are activating a single synapse; or, given the relatively sparse distribution of granule cell spines, that at least they're activating a single synapse onto a given granule cell.

I am still not clear why all the pharmacological manipulations decreased sIPSC amplitude, but the authors have done the rather simple control experiment suggested by reviewer 2, and I am not sure that this point invalidates their results or their interpretation. I suggest the authors note this issue explicitly in the Discussion and that we give them the benefit of the doubt.

---

## [Author Response]

[Editors’ note: the authors resubmitted a revised version of the paper for consideration. What follows is the authors’ response to the first round of review.]

Reviewer #1:Lage-Rupprecht and colleagues describe biophysical mechanisms of MC-GC recurrent inhibition using minimal stimulation of GC spines. They use a local two-photon uncaging of glutamate to evoke IPSCs that are smaller than spontaneous events. Surprisingly for a minimal stimulation of this magnitude, they observe that voltage gated Na channels, high voltage activated Ca channels as well as NMDA receptors are all necessary to evoke a recurrent IPSC. It should be noted that unlike previous studies in the field, the authors demonstrate the NMDA requirement under a physiological condition. Using an ultrastructural evidence, they demonstrate that NMDA receptors are strategically located i.e., in proximity to the release sites on the GC spine, well within the 100 nm boundary that theoretical studies suggest is crucial if the calcium influx lead directly to vesicle release. Compartmentalised Ca influx via a local NMDA activation would be an intriguing and ideal mechanism to implement specific yet flexible connectivity between MCs and GCs.The data are of high quality, and the insights on the NMDA receptor involvement is new and very important. There is some confusion as to whether or not this is a study about a single-spine stimulation, while the majority of the discussion is spent on co-operative activation by several MCs. While it is not necessary that the data deals with a single-spine stimulation strictly, a manuscript may improve by clarifying this point. These are elaborated below in interrelated points.

We agree that this issue is potentially confusing and have reduced the respective parts of the Discussion, also in response to similar comments by reviewer 2.

1) Regarding the minimal stimulation used, the intention and interpretation seems to be that the two-photon uncasing leads to a single-spine stimulation. It is not clear that this is demonstrated. For example, in Figure 1X, the amplitude of IPSC shows some fluctuations. Does a histogram show a quantal nature or at least is the fluctuation consistent with a single contact?

This is an intriguing point, also since I have some prior experience in studying the properties of unitary synaptic transmission (Feldmeyer, Egger, Sakmann 1999). Unfortunately, because of the high spontaneous activity background and the low apparent release probability, in most experiments the number of reliable measurements of single responses is too low to allow to compute such histograms. For our main analyses of pharmacological effects we always used amplitudes measured from averaged responses. However, I have rechecked our set of stability experiments where 10 trials were followed by an interval of 10 minutes and then another 10 trials (to mimick the time course of pharmacological manipulations). Although the number of averaged responses per interval were low (average n = 2.8 ± 1.1, now included in the manuscript), the amplitudes of the averaged were very similar for the two intervals as can be seen from Figure 1F. While this observation is not a firm proof of quantal release, I would argue that it is at least in line with it, since a connection with two or three release sites instead of one would show a considerably higher CV and thus fluctuations between the average amplitudes are more likely to occur, since these are based on a low sampling number. On the other hand, the low Pr will render it fairly unlikely to observe double release events.

We also conducted additional experiments to further ensure single-spine activation, at least with regard to the same GC – see response to point 3.

2) According to a recent BioRxiv preprint (https://doi.org/10.1101/2020.01.10.901397 ), where some of the current authors describe Ca^2+^ dynamics in GC dendrites following a simultaneous uncaging of glutamate with incremental number of stimulated sites, demonstrating that Nav, HVACC, and NMDA receptors are recruited when several GC spines are stimulated, but not 1 spine. This may suggest the conditions used in the current manuscript may activate several spines. This may occur, for example, if more than 1 dendro-dendritic contacts exist between each GC-recorded MC pair, and clamping the recorded MC voltage at +10 mV (lower than the reversal potential of Ca^2+^ and high enough to activate the channel) could lead to some glutamate release on other parts of the GC dendrite. Have the authors tried to record at a more hyperpolarized voltage (maybe with a different internal solution to enhance the GABAergic conductance)?

This preprint is by now accepted by PLoS Biology and available online at their website PLoS Biol 18(9): e3000873. https://doi.org/10.1371/journal.pbio.3000873. We wish to clarify that the mentioned recruitment of active conductances in this study is referring to additional recruitment during activation of more than one spine, as also discussed by the reviewer in the next point. All these conductances are also already recruited for single spine activation within the activated spine, as described in Bywalez et al., 2015 and stated in the manuscript, both in the Introduction for Nav and HVACC and also in the Discussion.

The current manuscript also contains a set of experiments conducted with a different internal at – 70 mV, which was our initial recording configuration (Materials and methods). We switched to +10 mV in the interest of further noise reduction. There was no difference between the two conditions with respect to the effect of TTX. As to the concern that glutamate release from MCs held at + 10 mV might have confounded the results, we wish to point out that the clamp to +10 mV was performed at least 10 minutes before the beginning of uncaging, as also indicated in the Materials and methods.

3) Alternatively, it is possible that the apparent discrepancy above (between this and the previous studies) simply reflects a difference in the anatomical scale of analysis. That is, it is possible that the active conductance was already present with 1 stimulation site in the GC study, but highly localized within the spine head volume. So, an analysis of the spatial distribution of evoked calcium signal in GC spines may resolve this issue (if the data in the BioRxiv paper can be analyzed this way).

This is in line with our understanding, and as outlined above we have further clarified the manuscript in this respect. As to the spatial distribution, we have now conducted extra experiments showing not only that Ca^2+^signals do not spread into the dendrite but that neighboring spines also do not sense the uncaged glutamate (at least within our detection thresholds). See response to 4.

The experiments in the PLoS Biology paper show the very same result, however we did not wish to use this result here since in that study uncaging spots were generated via a spatial light modulator, which might cause a larger uncaging volume in the z-dimension than the conventional uncaging used here, thus glutamate could be more likely to spread to adjacent spines (even though this is not backed by our observations).

4) If indeed it turns out that the minimal stimulation involved a recruitment of several spines, it would be interesting to see if, in the Ca signals in the GC dendrite, the NMDA-receptor mediated calcium entry is confined to the activated spine (e.g., analysis of calcium signals in stimulated spine vs. non-stimulated spine next to it). This may support their idea of localized release during a more global event, such as during the co-operative activation by multiple MCs, as depicted in the schematics and elaborated in the Discussion.

We have now performed this experiment in a set of 11 spine pairs (Figure 1—figure supplement 1C,D,E). Due to the low spine density of GCs (approx 2 per 10 µm) and the long spine necks, it was not easy to find two nearby spine heads in the same focal plane. We could not find any evidence for a spillover of glutamate to non-stimulated spines on the same cell. Even though it is deemed unlikely that a given MC is contacted by more than one spine from the same GC (Woolf et al., 1994).

5) Overall, the Discussion is quite lengthy and somewhat removed from the data currently presented. Please edit accordingly.

We agree, we have cut down a lot of the Discussion, especially the part referring to stronger activation of GCs and lateral inhibition. These ideas are now described in a rather succinct manner; the extended version will be incorporated in a review that we are working on.

Reviewer #2:This manuscript uses glutamate uncaging, whole cell recordings, and pharmacology to dissect the role of different currents in reciprocal inhibition at dendro-dendritic MC-GC synapses. Specifically, the authors record whole cell currents from MCs while performing glutamate uncaging onto nearby GCs and observe both spontaneous and evoked IPSCs in MC held at a depolarized potential. They then apply antagonists of GABA receptors, Nav channels, HVACCs, and NMDARs and show that in all cases both spontaneous IPSCs and evoked IPSCs are reduced. The results are interpreted as support for the "mini-neuron" hypothesis, in which single GC spines act as neurons with spike generation leading to HVACC activation and release. Further, the contribution of NMDARs to this process is interpreted as evidence for "on-demand" lateral inhibition between co-active columns.The experiments shown appear to have been done carefully and the ideas proposed are interesting. However, I do think some of the results are over-interpreted here, given the strong suppression of spontaneous IPSCs observed with all pharmacological manipulations. The interesting ideas put forward in the lengthy Discussion might be effectively paired with a computational model to make an effective case in a manuscript of their own.Major comments:1) Overall the manuscript is written assuming a close familiarity with the prior literature on this circuit and synapses, and in particular with a previous manuscript from this lab. References to "the previous manuscript" are made frequently in the text without an exposition of what is shown there. The manuscript should be written as a stand-alone document that provides the necessary background for a reader to understand and interpret the findings as is.In the revised version we have now always explicitly mentioned the relevant results ofBywalez et al.2) All of the pharmacological manipulations shown decrease spontaneous IPSCs as well as IPSCs evoked by uncaging. Therefore, it is not clear of the mechanisms are specific to spine-evoked release or simply represent an overall decrease in excitability of the GC or network. At the very least evoked IPSC amplitude should be compared to baseline both before and after application of drugs.

The spontaneous IPSC amplitude data had been analyzed, and we apologize for not having included them in the original manuscript, since indeed this is a critical point. For none of the three drugs (TTX, CTX, APV) there was a systematic reduction of sIPSC amplitude, rendering a postsynaptic effect rather unlikely. Even though sIPSC amplitudes were never reduced on average, we also now checked whether there might be a correlation between the strengths of the drug effects on urIPSC amplitudes and on sIPSC frequencies. No such positive correlation was observed for any of the drugs (S2C, S3C, S4C), arguing against network effects interfering with urIPSCs. All these data are now included also in the manuscript (Results, Discussion). We have added supplementary figures to all three pharmacological manipulation figures (Figure 2—figure supplement 1, Figure 3—figure supplement 1, Figure 4—figure supplement 1) that provide the mentioned observations on spontaneous activity; the panels on sIPSC frequencies have been moved into these figures.

3) The spatial extent of excitation produced by glutamate uncaging on GCs is not measured experimentally. While this excitation is assumed to be confined to a single spine this is never shown and seems to be critical to the interpretation. Voltage or Ca imaging from GCs during uncaging would be important to show this.

This is an important issue that has been improved on by additional experiments. While we have shown previously, that ∆Ca^2+^ does not spread to the adjacent dendrite and these data are now reviewed more carefully in the current manuscript, we now provide evidence that TPU also does not cause excitation of neighboring spines, investigating 11 spine pairs. These data are now shown in Figure 1—figure supplement 1.

See also response to reviewer 1 point 4.

4) The Discussion is very long and puts forward a number of hypotheses that are significant extrapolations from the presented data. The ideas put forward here are quite interesting but I think would be more compelling if supported by a computational model and could easily constitute a manuscript of their own coupled with the extensive literature review that supports the model proposed.

This was also criticized by reviewer 1. We are already engaged in the first steps of such a modelling study and are planning to outline the hypotheses in a conceptual review format, hoping to elicit interest in these ideas in the community.

[Editors’ note: what follows is the authors’ response to the second round of review.]

Reviewer #2:The authors performed a series of high-quality experiments that make a strong argument for the mini-neuron model, with an interesting interpretation of these data that suggests that granule-cell-mediated lateral inhibition is specific to coactive glomerular columns.Regarding the first major concern that they may be activating multiple spines, they now provide additional data that show only a small fraction of responses with blind uncaging, which is nice. However, more compellingly, when they do calcium imaging on two neighboring spines, they only see transients in the stimulated spine and not in its neighbor. Together, these data strengthen their argument that they are activating a single synapse; or, given the relatively sparse distribution of granule cell spines, that at least they're activating a single synapse onto a given granule cell.I am still not clear why all the pharmacological manipulations decreased sIPSC amplitude, but the authors have done the rather simple control experiment suggested by reviewer 2, and I am not sure that this point invalidates their results or their interpretation. I suggest the authors note this issue explicitly in the Discussion and that we give them the benefit of the doubt.

This seems to be a partial mixup since all the manipulations did NOT decrease sIPSC amplitudes (although all of them decreased sIPSC frequencies). While this finding was not mentioned in the very first version of the manuscript, the amplitude data have been included in the revision, along with proof that there is no correlation between the frequency effects and the effect on urIPSCs. We also discuss possible reasons and findings from other labs with regard to the decrease in frequency in greater detail, which are actually in support of our observation. In any case, we now have included an explicit statement of the potentially problematic observation that both evoked and spontaneous activity were reduced by the pharmacological manipulations in the Discussion:

“All three antagonists of urIPSC generation (TTX, CTX, APV) were found to also substantially reduce sIPSC frequency, which might imply that the reductions of urISPCs are due to a general reduction in excitability or network effects. However, none of the antagonists affected sIPSC amplitude or reduced sIPSC frequency in a manner correlated with their effect on evoked urISPCs, and similar effects on spontaneous activity were also reported elsewhere.”

In the meantime, I consider it also likely based on unpublished observations that the effect of APV on spontaneous activity could have been further enhanced by the presence of DNI, which enhances spontaneous activity in itself (Figure 1—figure supplement 1B) and thus might increase the amount of ambient glutamate. This possibility has now been added to the Discussion:

“(4) Finally, it is conceivable that the enhancement of spontaneous activity by the caged compound (Figure 1—figure supplement 1B) results in slightly elevated levels of ambient glutamate and thereby further augments the contribution of NMDARs to spontaneous activity by the three aforementioned mechanisms.”